



# Dissolved Organic Matter at the Fluvial-Marine Transition in the Laptev Sea Using in situ Data and Ocean Color Remote Sensing

Bennet Juhls[1], Pier Paul Overduin[2], Jens Hölemann[3], Martin Hieronymi[4], Atsushi Matsuoka[5], Birgit Heim[2], Jürgen Fischer[1]

[1]Institute for Space Sciences, Department of Earth Sciences, Freie Universität Berlin, Berlin, Germany
[2]Alfred Wegener Institute Helmholtz Centre for Polar and Marine Research, Potsdam, Germany
[3]Alfred Wegener Institute Helmholtz Centre for Polar and Marine Research, Bremerhaven, Germany
[4]Institute of Coastal Research, Helmholtz-Zentrum Geesthacht, Geesthacht, Germany
[5]Takuvik Joint International Laboratory, Département de Biologie, Université Laval, Canada

*Correspondence to*: Bennet Juhls (bjuhls@wew.fu-berlin.de)

**Abstract.** River water is the main source of dissolved organic carbon (DOC) in the Arctic Ocean. DOC plays an important role in the Arctic carbon cycle and its export from land to sea is expected to increase as ongoing climate change accelerates permafrost thaw. However, transport pathways and transformation of DOC in the land-to-ocean transition are mostly unknown. We collected DOC and $a_{CDOM}(\lambda)$ samples from 11 expeditions to river, coastal and offshore waters and present a new DOC-$a_{CDOM}(\lambda)$ model for the fluvial-marine transition zone in the Laptev Sea The $a_{CDOM}(\lambda)$ characteristics revealed that the DOM in samples of this dataset are primarily of terrigenous origin. Observed changes in $a_{CDOM}$ and its spectral slopes indicate that DOM is modified by microbial- and photo-degradation. Ocean Color Remote Sensing (OCRS) provides the absorption coefficient of colored dissolved organic matter ($a_{CDOM}(\lambda)^{sat}$) at $\lambda=440$ or 443 nm, which can be used to estimate DOC concentration at high temporal and spatial resolution over large regions. We tested the statistical performance of five OCRS algorithms and evaluated the plausibility of the spatial distribution of derived $a_{CDOM}(\lambda)^{sat}$. The ONNS algorithm showed the best performance compared to in situ $a_{CDOM}(440)$ ($r^2=0.72$). Additionally, we found ONNS-derived $a_{CDOM}(440)$, in contrast to other algorithms, to be partly independent of sediment concentration, making ONNS the most suitable $a_{CDOM}(\lambda)^{sat}$ algorithm for the Laptev Sea region. The DOC-$a_{CDOM}(\lambda)$ model was applied to ONNS-derived $a_{CDOM}(440)$ and retrieved DOC concentration maps showed moderate agreement to in situ data ($r^2=0.53$). The in situ and satellite-retrieved data were offset by up to several days, which may partly explain the weak correlation for this dynamic region. Satellite-derived surface water DOC concentration maps from MERIS satellite data demonstrate rapid removal of DOC within short time periods in coastal waters of the Laptev Sea, which is likely caused by physical mixing and different types of degradation processes. Using samples from all occurring water types leads to a more robust DOC-$a_{CDOM}(\lambda)$ model for the retrievals of DOC in Arctic shelf and river waters.



## 1 Introduction

Large volumes of fresh water and dissolved organic matter (DOM) are discharged by Arctic rivers into the Arctic Ocean (Cooper et al., 2005; Dittmar and Kattner, 2003; Stedmon et al., 2011). Recent studies predict an increase of DOM flux to the Arctic Ocean with continued climate warming and permafrost thawing (Camill, 2005; Freeman et al., 2001; Frey and

Smith, 2005). This will lead to a cascade of effects on the physical, chemical and biological environment of Arctic shelf waters (Stedmon et al., 2011). These include an increase of radiative heat transfer into surface waters, changes in carbon sequestration, and reductions of sea-ice extent and thickness (Hill, 2008; Matsuoka et al., 2011).

The Laptev Sea is a wide shelf sea in the eastern Arctic characterized by fresh surface waters from the Lena River, which delivers around one fifth of all river water to the Arctic Ocean. River water is the main source of DOM and thus of

dissolved organic carbon (DOC) and colored dissolved organic matter (CDOM) to the Laptev Sea shelf (Cauwet and Sidorov, 1996; Gonçalves-Araujo et al., 2015; Kattner et al., 1999; Lobbes et al., 2000; Thibodeau et al., 2014; Vantrepotte et al., 2015). Moreover, the Lena River has the highest peak concentrations of DOC of all Arctic rivers. The fate and transformation of DOM as it is discharged to the Arctic Ocean, however, are not well known. Physical and biological processes, such as photodegradation (Gonçalves-Araujo et al., 2015; Helms et al., 2008, 2014; Opsahl and Benner, 1997) and

microbial degradation (Benner and Kaiser, 2011; Fasching et al., 2015; Fichot and Benner, 2014; Matsuoka et al., 2012, 2015), as well as mineralization (Kaiser et al., 2017) and flocculation (Asmala et al., 2014; Guo et al., 2007), are responsible for the modification and removal of DOM from river-influenced surface waters. Given the strong seasonality of Lena River runoff (Yang et al., 2002), DOC concentration varies greatly in time and space (Amon et al., 2012; Cauwet and Sidorov, 1996; Raymond et al., 2007; Stedmon et al., 2011). Once exported to the sea, rapid transport of water masses and dislocation

of fronts cause rapid changes in concentrations of surface water constituents at any given location.

Therefore, DOC sampling at high temporal and spatial resolutions over long periods is necessary to understand these changes. Discrete in situ sampling of DOC during expeditions provides point measurements at the time of sampling and is complicated by the difficulty of accessing shallow water for ocean-going vessels. The resulting inadequacy of sample coverage in space and time can be overcome by using Ocean Color Remote Sensing (OCRS) data. The absorption coefficient

of CDOM ($a_{CDOM}(\lambda)$) at a reference wavelength $\lambda$ (usually $\lambda$=443 nm or $\lambda$=440 nm is used) is an optical property of the water and can also be derived with OCRS during ice and cloud-free periods. Hereinafter, we refer to satellite derived $a_{CDOM}(\lambda)$ as $a_{CDOM}(\lambda)^{sat}$. CDOM absorbs light in the ultraviolet and visible wavelengths (Green and Blough, 1994) and can be used to estimate DOC concentration. Thus, OCRS provides an alternative to discrete water sampling (Hansell et al., 2002; Matsuoka et al., 2017). DOC concentration maps with high spatial and temporal resolution will improve our understanding of DOC

dynamics in fluvial-marine transition zones and better quantify carbon cycling. However, most OCRS retrieval algorithms have focused on optically deep (Case 1) waters, which usually correspond to open ocean where all optical water constituents are coupled to chlorophyll concentration (Kutser et al., 2017; Mobley et al., 2004). Generally, the Laptev Sea coastal to central-shelf waters and Lena River water can be classified as extreme-absorbing and high-scattering waters with high





optical complexity (Case 2) (Heim et al., 2014; Hieronymi et al., 2016). Algorithms for Case 1 water do not provide reasonable estimates of water constituents in optically complex waters (Antoine et al., 2013). Hieronymi et al. (2017) use a novel algorithm for the retrieval of OCRS products such as $a_{CDOM}(440)$. This algorithm is specifically designed for a broad range of concentrations of different water constituents including extremely high absorbing waters with $a_{CDOM}(440)$ of up to

$20\ m^{-1}$.

In order to estimate DOC concentration from $a_{CDOM}(\lambda)$, a number of empirical relationships between in situ DOC and $a_{CDOM}(\lambda)$ for Arctic regions are presented in recent studies (Fichot and Benner, 2011; Gonçalves-Araujo et al., 2015; Mann et al., 2016; Matsuoka et al., 2012; Örek et al., 2013; Spencer et al., 2009; Walker et al., 2013). However, the DOC-$a_{CDOM}(\lambda)$ relationship can vary in different water types and can change between seasons and regions (Mannino et al., 2008;

Vantrepotte et al., 2015). Furthermore, existing Arctic datasets of DOC and $a_{CDOM}(\lambda)$ taken in situ are almost all limited to either offshore, coastal or river waters, so that a DOC-$a_{CDOM}(\lambda)$ relationship has not been established for the range of water types in Arctic coastal waters. Samples from near-shore waters from Arctic shelves are under-represented in these datasets. In order to obtain synoptic DOC concentration maps that cover the fluvial-marine transition, a relationship valid for a combination of these different water types is required.

Spectral shapes of $a_{CDOM}(\lambda)$ can provide additional information on the DOM quality and about involved biogeochemical processes that modify the DOM (Carder et al., 1989; Matsuoka et al., 2012; Nelson et al., 2004, 2007). Various studies use the $a_{CDOM}(\lambda)$ slope in the UV domain (S275-295) as an indicator of the photodegradation history of the $a_{CDOM}(\lambda)$ (Fichot and Benner, 2012; Helms et al., 2008; Del Vecchio and Blough, 2002). Spectral characteristics of $a_{CDOM}(\lambda)$ and their correlation to the DOC specific absorption coefficient ($a^*_{CDOM}(440)$) vary across the Eastern Arctic Ocean (EAO)

and the Western Arctic Ocean (WAO) (Stedmon et al., 2011).

In this study we aim to better understand the transport of organic material from land to sea in the Arctic and improve its detection at regional scale in the Laptev Sea, where the Lena River provides a major source of DOM to the Arctic Ocean. For this, we compile a dataset of DOC and $a_{CDOM}(\lambda)$ samples collected during multiple expeditions to the Laptev Sea and Lena River region in order to investigate the optical characteristics and variability of $a_{CDOM}(\lambda)$. With this

dataset. we develop a new DOC-$a_{CDOM}(\lambda)$ relationship which we apply to OCRS data in order to estimate DOC concentration from space. We test and evaluate the accuracy of different OCRS algorithms for the fluvial-marine transition zone in the Laptev Sea.

## 2 Material and Methods

### 2.1 Study area & expeditions

The in situ data presented in this study are compiled from several, mostly unpublished datasets from Russian-German expeditions to the Lena River and Laptev Sea that took place from 2010 to 2017 (Table 2). Sampling locations of this dataset



include large parts of the western and central Laptev Sea shelf, coastal regions around the Lena River Delta and channels of the Lena River (Fig. 1).

All ship- and land-based sampling took place during the ice-free period between the end of June and mid-September. Only one land-based sampling in the central Lena River Delta took place between the end of May and the end of June, during Lena River peak discharge after the ice break-up. The ship expeditions, which covered offshore shelf waters (NE10, YS11, VB13 and VB14), were conducted on board RV *Nikolay Evgenov* (NE), RV *Jacob Smirnitsky* (YS) and RV *Viktor Buynitskiy* (VB), respectively. For the other ship expeditions, smaller boats were used for sampling in coastal waters or on the Lena River. Water sampling at the research station on Samoylov Island (LD14) was carried out from small boats or from the shore (Fig. 1). Table 1 shows a summary of sampling periods, water types and the sampled parameters of the individual expedition datasets.

## 2.2 Hydrographic characteristics and sample processing

For each sampling location included in this dataset, vertical profiles of the water column temperature and salinity were measured with a CTD (Sontek CastAway CTD for LD14, LD15, LD16, Byk17 and a Seabird 19+ for LD10, LD13, NE10, YS11, VB13, VB14, GA13). In this study we use the practical salinity unit (psu) to describe salinity. Aboard ships and boats, water samples were taken using Niskin bottles or an UWITEC water sampler at defined depths. Since this study focuses on improving satellite retrievals, only surface water samples (discrete samples from 2 and 5 m water depth) were included in the compiled dataset. Based on visual examination of the water column characteristics we also included samples from 10 m depth wherever a thick homogeneous upper mixed layer was present. During the expedition LD14, water samples were taken from the shore of Samoylov Island at around 0.5 m depth using 5-liter glass bottles

Water samples for DOC analysis were filtered through 0.7µm GF/F filter and acidified with 25 µL HCl suprapur (10 M) after sampling. Samples were stored cool and dark for transport. DOC concentrations were measured using high temperature catalytic oxidation (TOC-VCPH, Shimadzu). Three measurements of each sample were averaged and after each 10 samples, a blank and a standard (*Battle-02*, *Mauri-09* or *Super-05* certified reference material from National Laboratory for Environmental Testing, Canada) were measured for quality control.

Samples for $a_{CDOM}(\lambda)$ analysis were filtered through 0.22 µm Millipore GSWP filters (GA13, LD16, Byk17) or 0.7 µm Whatman GF/F (LD10, YS11, VB13, VB14, LD14, LD15) after sampling. 100 ml filtrate was stored cool and dark in amber glass bottles until further analysis. $a_{CDOM}(\lambda)$ was measured with a spectrophotometer (SPECORD 200, Analytik Jena) by measuring the absorbance ($A_\lambda$) at 1 nm intervals between 200 and 750 nm. Absorption was calculated from the resulting absorbance measurements via

$$a_{CDOM}(\lambda) = \frac{2.303*A_\lambda}{L}, \tag{1}$$

where L is the path length (length of cuvette), to calculate the $a_{CDOM}(\lambda)$. Fresh Milli-Q water was used as reference. The cuvette length varied depending on the expected absorption in the sampled water (1 or 5 cm for river or coastal waters, 5 or 10 cm for offshore shelf waters). Resulting $a_{CDOM}(\lambda)$ spectra were corrected for baseline offsets by subtracting the mean



absorption between 680 and 700 nm, assuming zero absorption at > 680 nm. We focussed on $a_{CDOM}$ at 443 nm since most OCRS algorithms use this wavelength to retrieve $a_{CDOM}(\lambda)$. This wavelength corresponds to one spectral band of most multispectral satellite sensors. Spectral slopes of $a_{CDOM}(\lambda)$ were calculated by non-linearly fitting the following equation:

$$a_{CDOM}(\lambda) = a_{CDOM}(\lambda_0) * e^{-S(\lambda - \lambda_0)}, \tag{2}$$

where $a_{CDOM}(\lambda_0)$ is the absorption coefficient at reference wavelength $\lambda_0$ and $S$ is the spectral slope of $a_{CDOM}(\lambda)$ for the chosen wavelength range. Spectral slopes of $a_{CDOM}(\lambda)$ were calculated fitting Eq. (2) for the individual wavelength range. The DOC specific absorption coefficient at $\lambda$=440 nm was calculated with $a^*_{CDOM}(440) = a_{CDOM}(440)/DOC$.

## 2.3 Satellite data

In order to monitor spatiotemporal variability of DOC in surface waters and test the applicability of the established DOC-
$a_{CDOM}(\lambda)$ model from this study, we used OCRS. We applied the DOC-$a_{CDOM}(\lambda)$ model to calculate DOC concentration from satellite-retrieved $a_{CDOM}(\lambda)$. For this study, we chose the Medium Resolution Imaging Spectrometer (MERIS) because of its high spectral resolution and spectroradiometric quality.  Many OCRS algorithms were developed specifically for this sensor and are designed for coastal waters. MERIS L1 satellite scenes at reduced resolutions (1 km spatial resolution) were obtained from the MERIS Catalogue and Inventory (MERCI). We checked all expedition periods for cloud-free MERIS satellites
scenes but only two expeditions from 2010 (LD10 and NE10 ship expeditions) could be used to evaluate the performance of the remote sensing retrieval of the surface water DOC concentration. During those periods, we identified a few scenes with substantial cloud-free data coverage that were acquired during the 2010 expedition periods. Table 1 lists MERIS scenes used in this study. In order to visualize satellite-derived results, we generated mosaic images containing the average of the overlap from multiple satellite scenes to extend the data coverage between cloud gaps. To compare in situ with satellite data, we used
extracted pixel values from each single image. To discuss processes that cause differences between satellite images we extracted reanalysis surface wind data (4 times daily) from the National Centers for Environmental Prediction.

Hieronymi et al. (2017) developed the OLCI (Sentinel-3 Ocean and Land Colour Instrument) Neural Network Swarm (ONNS) in-water algorithm for the retrieval of OCRS products, among them $a_{CDOM}(440)$. This algorithm is designed for broad concentration ranges of different water constituents, including extremely high absorbing waters. The algorithm
differentiates 13 optical water types (OWT) and uses specific neural networks (NN) for each OWT. Every NN is trained for narrow concentration ranges. The values of $a_{CDOM}(440)$ used for the training of the NN's are up 20 m$^{-1}$. The final product is a weighted sum of all NNs depending on OWT membership. The standard atmospheric correction of ONNS, namely the C2RCC (Brockmann et al., 2016), is applied. ONNS makes use of 11 out of the 21 OLCI bands, including the 400 nm band, which is the only one not delivered by MERIS. In order to retrieve OCRS products with ONNS from MERIS imagery, a
band adaptation NN-algorithm is utilized to extrapolate remote sensing reflectance at 400 nm, which is usually provided with an uncertainty <5 % for these waters (Hieronymi, 2019). Note that the ONNS-algorithm uses the $a_{CDOM}(\lambda)$ wavelength 440 nm whereas all other algorithms use 443 nm.




Additionally, we tested the following open source OCRS algorithms: (1) FUB/WeW MERIS Case-2 Water properties processor (FUB/WeW) (Schroeder and Schaale, 2005) developed for $a_{CDOM}(443)$ up to 1 m$^{-1}$), MERIS case 2 water algorithm (C2R) (Doerffer and Schiller, 2007) ($a_{CDOM}(443)$ up to 1 m$^{-1}$) which is used for the MERIS 3rd Reprocessing of ESA's distributed products, and the Case 2 Regional CoastColour (C2RCC) (Brockmann et al., 2016) with

C2RCC ($a_{CDOM}(443)$ up to 1 m$^{-1}$) and C2X ($a_{CDOM}(443)$ up to 60 m$^{-1}$). All algorithms used in this study use neural networks trained with databases of radiative transfer simulations or in situ measurements or both to invert the satellite signal into inherent optical water properties. In this study the atmospheric correction from Case 2 Regional CoastColour processor (C2RCC) (Brockmann et al., 2016) was used to provide atmosphere corrected reflectances for the OCRS algorithms ONNS, C2R, C2RCC and C2X. For the FUB/WeW algorithm the atmospheric correction provided by the FUB/WeW processor

(Schroeder and Schaale, 2005) was used.

### 2.3.1 Functions for satellite retrieval evaluation

In order to evaluate the retrieval of a $a_{CDOM}(\lambda)^{sat}$ from the tested OCRS algorithms, we used a number of evaluation parameters suggested by Bailey and Werdell, (2006) and Matsuoka et al., (2017). Among them, we use the median of satellite to in situ ratio (Rt), the semi-interquartile range (SIQR), the median absolute percent error (MPE), and root mean

square error (RMSE). The evaluation parameters are defined as follows:

$$Rt = median(\frac{X_{sat}}{X_{in\ situ}}), \qquad (3)$$

$$SIQR = \frac{Q_3 - Q_1}{2}, \qquad (4)$$

$$MPE = median(100 * \left|\frac{X_{sat} - X_{in\ situ}}{X_{in\ situ}}\right|), \qquad (5)$$

$$RMSE = \sqrt{\frac{\sum_{n=1}^{N}[X_{sat} - X_{in\ situ}]^2}{N}}, \qquad (6)$$

where $X_{sat}$ and $X_{in\ situ}$ are the satellite-derived and in situ measured $a_{CDOM}(443)$, respectively. Q1 represents the 25$^{th}$ ratio percentile and Q3 represent the 75$^{th}$ ratio percentile.

### 3 Results

### 3.1 Spatial and temporal variability of DOC and aCDOM

To examine variability of DOC and CDOM optical properties along the land-ocean continuum of the Lena-Laptev system,

we generated a large dataset that covers spring freshet through late summer from 2010 to 2017 (Table 2). Compared to previously published datasets (Gonçalves-Araujo et al., 2015; Mann et al., 2016; Matsuoka et al., 2012; Walker et al., 2013), this dataset compiles not only samples of one water type but covers river, coastal and offshore waters throughout the most variable portion of the open water season.



To better understand characteristics of DOC and $a_{CDOM}(\lambda)$ in freshwater-marine waters, the compiled dataset was first classified into three water types according to salinity: (1) fresh river water with salinities from 0-0.2, (2) mesohaline coastal water with salinities from 0.2-16 and (3) offshore waters with salinities >16.

Overall, DOC concentrations tended to decrease from river to offshore. The same trend was also observed in $a_{CDOM}(443)$. In river water, DOC concentrations and $a_{CDOM}(443)$ ranged from 370 to 1315 µmol L$^{-1}$ (median=779 µmol L$^{-1}$) and 1.17 to 7.91 m$^{-1}$ $a_{CDOM}(443)$ (median=3.61 m$^{-1}$), respectively (Fig. 2a and b). DOC concentrations and $a_{CDOM}(443)$ in coastal waters ranged from 205 to 923 µmol L$^{-1}$ (median=590 µmol L$^{-1}$) and 0.71 to 3.79 m$^{-1}$ (median=2.05 m$^{-1}$), respectively. Values in offshore water were the least variable of all three water types with DOC concentrations from 91 to 606 µmol L$^{-1}$ (median=234 µmol L$^{-1}$) and $a_{CDOM}(443)$ from to 0.077 to 1.86 m$^{-1}$ (median=0.5 m$^{-1}$). Generally, observed DOC and $a_{CDOM}(443)$ values were similar to reported findings from the Lena River and Laptev Sea regions (Amon et al., 2012; Cauwet and Sidorov, 1996; Gonçalves-Araujo et al., 2015; Heim et al., 2014; Raymond et al., 2007; Stedmon et al., 2011).

The spectral UV slope (S275-295) (Fig. 2c) showed similar maximum and median values for river (max.=0.0184 nm$^{-1}$, median=0.0155 nm$^{-1}$) and coastal waters (max.=0.0192 nm$^{-1}$, median=0.0161 nm$^{-1}$). We observed the lowest S275-295 in the Lena River water during the spring freshet at the beginning of June (LD14, Table 2). Offshore water has significantly higher S275-295 values ranging from 0.0158 to 0.0267 nm$^{-1}$ (median=0.195 nm$^{-1}$). For river and coastal water, S350-500 showed a similar variability as S275-295. The range of offshore water S350-500 however, showed substantially higher variation and covered a broad range (Fig. 2d).

In contrast to trends in DOC concentrations and $a_{CDOM}(443)$, $a_{CDOM}(\lambda)$ spectral slopes in two distinct spectral domain (S275-295 and S350-500) tended to increase from river to offshore. While the spectral slopes between river (max.=0.0184 nm$^{-1}$, median=0.0155 nm$^{-1}$) and coastal waters (max.=0.0184 nm$^{-1}$, median=0.0158 nm$^{-1}$) were not substantially different, those between the river and offshore were significantly different (p-value=< 10$^{-8}$).

**3.2 CDOM absorption characteristics**

We compared salinity and $a_{CDOM}(443)$ for the compiled dataset. As in other river-influenced waters, there was a tight correlation between $a_{CDOM}(443)$ and salinity, suggesting that physical mixing prevails and plays a role in near-conservative behavior of $a_{CDOM}(\lambda)$ (r²=0.87, n=283) (Fig. 3). For this analysis, only coastal and offshore waters were included since river water was constant in salinity but varied seasonally in $a_{CDOM}(443)$ (LD14, Table 2). In the coastal and offshore waters, $a_{CDOM}(443)$ decreased gradually with increasing salinity. The observed mixing line is similar to the reported mixing-line for Laptev Sea shelf waters from Heim et al., (2014), which was developed using parts of this compiled dataset (LD10 & YS11). The reported relationship from Matsuoka et al., (2012), however, shows generally lower $a_{CDOM}(443)$ values in waters of the WAO along the salinity gradient. S350-500 was very variable along the mixing-line. However, low $a_{CDOM}(443)$ along the mixing line had high S350-500 and higher $a_{CDOM}(443)$ had low S350-500.

Bulk information, combined use of magnitudes and spectral slopes of CDOM absorption are useful for understanding sources and/or processes involved in the modification of dissolved organic matter (e.g. Fichot and Benner,





(2012) and Helms et al., (2008). The strongest correlation was observed between $a_{CDOM}(443)$ and the UV slope S275-295 (Fig.4a, r=-0.84). Similar strong correlations were reported by Fichot and Benner, (2011) between $a_{CDOM}(350)$ and S275-295 for coastal waters of the Beaufort Sea in the WAO. Here, we used $a_{CDOM}(443)$ to make the findings useful for the OCRS community, which usually retrieves $a_{CDOM}$ at 443 nm. The spectral slope S350-500 showed moderate correlation with $a_{CDOM}(443)$ (Fig.4b, r=-0.54). Furthermore, a high number of S350-500 values were located outside the range of observed S350-500 values for coastal waters of the western Arctic (dashed lines, Fig. 4b).

We observed a moderate correlation between $a^*_{CDOM}(440)$ and S350-500 (Fig. 4c, r=-0.56). Most samples from this study are located below the $a^*_{CDOM}(440)$ limits of oceanic water reported by Nelson and Siegel, (2002) indicating that water samples from this study are primarily river influenced with higher aromaticity. The reported relationship between $a^*_{CDOM}(440)$ and S350-500 from (Matsuoka et al., 2012) deviates strongly in slope of the regression and range of $a^*_{CDOM}(440)$ values from this data from the fluvial-marine transition zone of the Laptev Sea.

### 3.3 DOC – CDOM relationship

Generally, retrieval of optical water properties and water constituents such as DOC from satellite data consists of three steps: (1) atmospheric correction of the top of atmosphere radiance to the water-leaving or the in-water reflectance, which is needed as input for the OCRS algorithms, (2) the retrieval of $a_{CDOM}(\lambda)^{sat}$ from the atmospherically-corrected reflectance received by satellite, and (3) if $a_{CDOM}(\lambda)^{sat}$ is retrieved from OCRS, DOC can be calculated using an in situ DOC versus in situ $a_{CDOM}(\lambda)$ relationship. In the following, we present regional evaluations of a number of OCRS algorithms suitable for coastal water and an improvement of (3) with a new DOC-$a_{CDOM}(\lambda)$ relationship from our compiled in situ dataset. The direct validation and evaluation of different atmospheric corrections (1) is beyond the scope of this study.

We observed a strong relationship between $a_{CDOM}(443)$ and DOC concentration for all water samples including river to marine waters (Fig. 5). One order of magnitude variation in DOC corresponded to more than 2 orders of magnitude of variation in $a_{CDOM}(443)$ for this sample set, and corresponded to the range from moderately absorbing waters (0.1 – 1.0 m⁻¹) to highly absorbing waters (>1.0 m⁻¹). After testing different regression models, the best fit was derived with a power function (Eq. (7), red line in Fig. 5):

$$DOC \ (\mu mol * L^{-1}) = 10^{2.525} * a_{CDOM}(443)^{0.659} \qquad (7)$$

The agreement between model and data (r²=0.96, n=227) allowed estimation of DOC by $a_{CDOM}(443)$ within a 50 % error range. The highest deviations from the fitted line corresponded to the transition zone between offshore shelf waters and coastal waters ($a_{CDOM}(443)$=0.5 – 1.5 m⁻¹) and to the very low end of the $a_{CDOM}(443)$ range (<0.5 m⁻¹). It is noted that the fitting model of this dataset using only offshore or river water samples would result in a lower slope (exponent=0.617 for coastal and offshore water, 0.606 for offshore water only) in the resulting DOC-$a_{CDOM}(443)$ model. Including coastal and river samples substantially increased the slope of the fit, which results in higher DOC estimates for high $a_{CDOM}(443)$. The reported relationship from Mann et al., (2016) is similar for high-$a_{CDOM}(443)$ river water but deviates for low-$a_{CDOM}(443)$





river water and coastal and offshore water. The model presented by Matsuoka et al., (2017) (blue line in Fig. 5) has a lower slope and results in highest differences for DOC estimation at high $a_{CDOM}(443)$.

Model coefficients for other selected $a_{CDOM}(\lambda)$ wavelengths are presented in Table A1 (Appendix A). Furthermore, the relationship between S275-295 and DOC had a slightly weaker correlation with DOC ($r^2$=0.92) than $a_{CDOM}(443)$.

## 3.4 Satellite retrieved CDOM

To estimate the surface water DOC concentration with the presented DOC-$a_{CDOM}(\lambda)$ model (Eq. (7), Fig. 5) and generate DOC concentration maps for large scales, we need a robust and accurate retrieval of $a_{CDOM}(\lambda)^{sat}$.

We examined the performance of five OCRS algorithm in terms of $a_{CDOM}(443)^{sat}$ retrieval using Eq. (3) to (6). For this purpose, $a_{CDOM}(443)^{sat}$ retrievals were compared to in situ data from within 10 days of the satellite retrievals. Our comparisons showed highly varying results (Fig. 6, Fig. B1 (Appendix B), Table 3) and strong under- or overestimation of $a_{CDOM}(\lambda)^{sat}$. Particularly in turbulent coastal waters, comparison of $a_{CDOM}(443)^{sat}$ with in situ $a_{CDOM}(443)$ is challenging, given the fact that the magnitude of CDOM absorption can vary greatly over a short time for the location of a given pixel. ONNS-derived $a_{CDOM}(\lambda)^{sat}$ performed best ($r^2$=0.716, Rt= 0.679, SIQR=0.217, %MPE=58.39, RMSE=0.436). The C2X algorithms performed similarly with a lower $r^2$ (0.65) and substantially higher %MPE (100.0) and RMSE (0.919). In addition to the comparison with in situ data, we evaluated the plausibility of the resulting spatial distributions and observed extremely high C2X-derived $a_{CDOM}(443)^{sat}$ values in the Lena River mouth (up to 10 m$^{-1}$). Such values of $a_{CDOM}(443)$ were not confirmed by any reported in situ data. ONNS-derived $a_{CDOM}(443)^{sat}$ showed values which are in the range of in situ observed $a_{CDOM}(443)$. Other algorithms show clear underestimations of $a_{CDOM}(443)^{sat}$ compared to in situ data (Fig.11, A2). Thus, ONNS was the only algorithm that produced $a_{CDOM}(\lambda)$ values in a similar range to in situ measured $a_{CDOM}(440)$.

## 3.5 Surface water DOC concentrations in coastal waters of the Laptev Sea

Using the presented DOC-$a_{CDOM}(\lambda)$ model, we generated satellite-derived images of surface water DOC concentrations for the Lena-Laptev Sea region. All scenes were processed with the ONNS algorithm and $a_{CDOM}(440)$ was averaged for each mosaic. DOC concentrations for two mosaics (Fig. 7b & 7d) were calculated by using the adapted model from Eq. (7) with coefficients for $a_{CDOM}(440)$ instead of $a_{CDOM}(443)$. The mean time difference between the two mosaics is 31 days. The DOC mosaic from early August 2010 (Fig. 7b) shows high DOC concentrations over large areas in the eastern Laptev Sea. Concentrations (>600 µmol L$^{-1}$) were highest in Buor Khaya Bay east of the Lena River Delta where the Lena River exports most of its water. The plume of the Lena River with high DOC concentrations (~500 µmol L$^{-1}$) had propagated northeastward in this scene. The DOC mosaic from September 2010 (Fig. 7d) shows generally lower DOC concentrations compared to the earlier scene. Highest concentrations were found in the coastal areas in Buor Khaya Bay (east of the Lena River Delta) and around the Olenek River Delta (west of the Lena River Delta) to the west of the Lena Delta. While ONNS performs well in river-influenced waters, we note that DOC concentrations at the northern edge can be influenced by cloud masking (patches of high DOC concentrations shown in northeast corner of Fig. 7d).



Both quasi-true color satellite images (Fig. 7a & 7c) show sediment-rich, strongly backscattering waters around the Lena River Delta resulting from fluvial transport. In the satellite image from 07.09.2010 (Fig.7c) there is also a large strongly backscattering area in the eastern Laptev Sea, where resuspension events in shallow water (5-10 m) occurred between both acquisition periods (Fig. 1). These resuspension events are not visible in the calculated DOC concentration maps at right (Fig. 7d).

### 3.5.1 In situ DOC vs. remotely-sensed DOC

To evaluate the satellite-retrieved DOC concentrations, we compared in situ and ONNS-retrieved DOC concentrations using the presented DOC-$a_{CDOM}(\lambda)$ model (Fig. 8) and investigated the plausibility of the DOC value ranges and the derived spatial patterns (Fig. 7b &c). This evaluation revealed a moderate performance ($r^2$=0.53, slope=0.61) (Fig. 8) despite several days difference in sampling times between satellite and in situ sampling. This comparison constitutes an evaluation and not a direct validation of the method. The latter is hampered by the lack of matching data and the time offsets between satellite acquisition and in situ sampling dates. The DOC-$a_{CDOM}(\lambda)$ model presented in this study improved satellite-derived estimates of DOC concentration compared to estimates using the DOC-$a_{CDOM}(\lambda)$ relationship reported by Matsuoka 2017 ($r^2$=0.46, Fig. 8).

To spare in situ data for this performance test, data from LD10 was not used to develop the DOC-$a_{CDOM}(\lambda)$ model (Eq. (7)). The DOC concentrations for NE10 were calculated from in situ $a_{CDOM}(443)$ measurements using Eq. (7), since no in situ DOC measurements were taken on NE10. These in situ DOC concentrations are therefore not independent, but were derived from the DOC-$a_{CDOM}(443)$ relationship for the entire dataset. However, samples from NE10 were not used for the development of the DOC-$a_{CDOM}(\lambda)$ relationship since in situ DOC was missing. We use the data to test the DOC retrieval for a wide range of concentrations. Further validation of the DOC retrieval will require additional in situ datasets collected simultaneously with cloud-free, open-water remote sensing acquisitions by using the MERIS successor OLCI.

## 4 Discussion

### 4.1 Sources and modification of DOM in the fluvial-marine transition

Our results showed that $a_{CDOM}(443)$ decreases as a function of salinity, indicating that river water is the main source of CDOM on the Laptev Sea shelf waters and in coastal waters and thus that most CDOM is of terrigenous origin (Fig. 3). Despite the tight relationship, some data points deviated from the mixing line in the salinity range from 2 to 24. Deviations from the mixing line can result from combined physical, chemical, and biological processes that modify CDOM optical properties (Asmala et al., 2014; Matsuoka et al., 2015, 2017). Helms et al., (2008) and Matsuoka et al., (2012) suggested that higher $a_{CDOM}(443)$ and lower S350-500 can be used as a proxy indicating that microbial degradation is more important than photodegradation. Indeed, we observed higher $a_{CDOM}(443)$ associated with lower S350-500 within a similar salinity range (Fig. 3), pointing towards stronger microbial degradation than photodegradation.



Flocculation can also modify CDOM optical properties by removing larger molecules once the river water encounters saline water. However, given the fact that this process occurs at low salinities (0 to 3; Asmala et al., (2014)), flocculation alone cannot explain the deviation of $a_{CDOM}$(443) values apart from the mixing line.

S275-295 was strongly correlated with $a_{CDOM}$(443) (Fig. 4a), which is mainly associated with the high content lignin chromophores in our samples (Fichot et al., 2013) and is partly explained by the long exposure of DOM to solar radiation and the resulting photodegradation (Hansen et al., 2016; Helms et al., 2008). Lena River water shows high lignin content and higher proportion of syringyl and vanyl phenols relative to p-hydrox phenols (Amon et al., 2012). Benner and Kaiser, (2011) showed that this could make the DOM more subject to photodegradation, which might supports why such a high correlation was observed.

Compared to the strong relationship between S275-295 and $a_{CDOM}$(443), a moderate correlation was observed for S350-500 versus $a_{CDOM}$(443) relationship (Fig. 4b), suggesting different degradation mechanisms were involved during the transition from river to coastal and offshore waters. Here we use S350-500, which is often used in the OCRS community, instead of S350-400, which is the wavelength range suggested by Helms et al., (2008). Note that the correlation between S350-500 and S350-400 is very high (r=0.98) and thus both slopes can be used. The mean river endmember value of S350-500 at salinity zero was 0.0163 nm$^{-1}$. This value tends to be lower in the EOA (including Lena river mouth) than in the WAO (Matsuoka et al., 2017; Stedmon et al., 2011). The higher $a_{CDOM}$(443) associated with the lower spectral slope observed in our river and coastal waters suggested more aromaticity in waters obtained from Lena-Laptev region (Stedmon et al., 2011). This was further demonstrated by our higher $a^{*}_{CDOM}$(443) (Fig. 4c).

The S350-500 versus $a_{CDOM}$(443) relationship showed a moderate but significant negative correlation and most samples were within a terrestrial range (dashed lines, Fig.4b). The fact that no samples were within the reported ranges of photodegradation for oceanic waters (solid lines, Fig. 4b) suggest that CDOM in coastal waters of the Laptev Sea would have been highly influenced by terrestrial inputs but with least photodegradation effect compared to that in oceanic waters (Matsuoka et al., 2015, 2017). It is likely that high turbidity and thus less transparent water of coastal regions in the Laptev Sea protects DOM from photodegradation. Data points outside of the ranges might indicate either microbial degradation and/or sea ice melt (Matsuoka et al., 2017). Given the only minor influence of ice melt waters during most of our field campaigns, microbial degradation is more likely for some of our samples, which is consistent with our explanation for deviated samples shown in Fig. 3.

The difference in optical properties of $a_{CDOM}(\lambda)$ observed between EAO and WAO is possibly caused by geological difference rather than climatic influences (Gordeev et al., 1996). This is partly supported by the chemical characterization of lignin phenols (Amon et al., 2012). Our results showed specificity of optical properties in the Lena and Laptev Sea and underline the necessity of discussing spectral optical properties when $a_{CDOM}$(443) and DOC concentration are estimated in this region.



## 4.2 Linking CDOM absorption to dissolved organic carbon concentration

Previous reported DOC-$a_{CDOM}(\lambda)$ models such as Walker et al., (2013), Örek et al., (2013) and Mann et al., (2016) for Arctic rivers, Matsuoka et al., (2012) for WAO and Gonçalves-Araujo et al., (2015) for coastal waters are restricted in their use to the water type of samples. Our presented DOC-$a_{CDOM}(\lambda)$ relationship improves reported models from Mann et al., (2016) and

Matsuoka et al., (2017) for the estimation of DOC from $a_{CDOM}(443)$ in DOC-rich waters in transition zones of river and seawater of the Lena-Laptev region. Matsuoka et al. (2017) provided satellite-retrieved DOC concentration maps for coastal waters of the Lena River Delta region, retrieved with a DOC-$a_{CDOM}(443)$ relationship developed using a pan-Arctic in situ dataset. However, the retrieved DOC concentrations were likely underestimated compared to in situ measurements in the coastal regions of the Laptev Sea presented in this study. In coastal or $a_{CDOM}(443)$-rich, river-influenced water, the difference

between the two relationships is expected to be highest. Applying the Matsuoka et al. (2017) relationship to $a_{CDOM}(443)$-rich waters outside its validity ranges (>3.3 $m^{-1}$) would result in underestimation of DOC compared to the relationship presented in this study. Taking mean Lena River $a_{CDOM}(443)$ of 4.1 $m^{-1}$, which is similar to the highest $a_{CDOM}(443)$ values in coastal waters, the difference in modelled DOC concentration between both relationships is 186.8 $\mu mol\ L^{-1}$. The main reason for this underestimation is likely the lack of near-coastal and river water samples with high DOC for the development of their

relationship. This hypothesis was confirmed by testing the relationship of our dataset by removing coastal and river water. This decreased the slope of the fitting model and lead to an underestimation of DOC in coastal and river waters (without coastal and river water: slope=0.617). The slope of the reported fitting model from Matsuoka 2017 (0.448) is lower compared to the fitting model from this study (all samples: slope=0.664). This difference highlights the importance of using a broad concentration range to develop such relationships.

The broad concentration range of the relationship presented here permits the generation of remotely sensed surface DOC concentration maps of the Laptev Sea across the fluvial-marine transition zone using $a_{CDOM}(443)$. The applicability of this relationship for other Arctic fluvial-marine transition regions (e.g. Yenisei, Ob, Kolyma, Mackenzie) is untested and the relationship may need to be extended with regionally specific data.

Previous studies using $a_{CDOM}(443)$ often focused on different wavelengths for $a_{CDOM}(\lambda)$. The shape of the DOC-

$a_{CDOM}(\lambda)$ relationship is strongly dependent on the chosen $a_{CDOM}(\lambda)$ wavelength: whereas DOC-$a_{CDOM}(350)$ shows a linear relationship, $a_{CDOM}(443)$ can be better described with a power function (see Eq. (7)). Table 4 (A1) provides coefficients for selected $a_{CDOM}(\lambda)$ wavelengths. We encourage the data publication of all available wavelengths for $a_{CDOM}(\lambda)$ measurements in future studies to enable direct comparisons between studies and regions.

### 4.2.1 ONNS-derived DOC

The evaluation of ONNS-derived $a_{CDOM}(\lambda)^{sat}$ performed best when tested with in situ data (Table 3). Thus, we selected the ONNS-derived $a_{CDOM}(440)^{sat}$ to calculate DOC concentration based on the Eq. (7). The evaluation of ONNS-derived DOC concentrations showed moderate performance ($r^2$=0.53). We suggest that the only moderate agreement likely results from




rapid movement of near-coastal water fronts. Fluvial-marine transition zones, as in this study area, are characterized by rapidly moving water fronts with large variations in DOC concentration. A spatial shift of a plume between in situ sampling and the satellite acquisition can cause large errors in the match-up performance. All samples from LD10 expedition are located in the near-coastal areas east of the Lena River Delta where rapid movements of water fronts are likely. This could

partly explain deviations comparing in situ measurements and the satellite derived DOC at a given pixel. Taking this into account, the observed agreement shows an adequate retrieval of DOC by satellite using the OCRS algorithm ONNS.

Using satellite-retrieved surface water DOC concentration maps (Fig. 7b & 7d), we demonstrated rapid changes in DOC concentrations in the Laptev over 1 month in late summer. The rapid DOC decrease can result from a combination of vertical mixing, dilution, and microbial- and photodegradation of the organic material in the surface water (Fichot et al.,

2013; Holmes et al., 2008; Mann et al., 2012). OCRS could potentially be used to monitor the rapid removal of DOM by degradation from surface waters on Arctic shelves.

## 4.3 OCRS algorithms in shallow Arctic fluvial-marine transition zones

We observed specific problems of $a_{CDOM}(\lambda)^{sat}$ retrievals for optically complex shallow shelf waters using different OCRS algorithms. The retrieved $a_{CDOM}(\lambda)^{sat}$ in shallow waters often shows a co-variation with high TSM, which is a result of high

particle backscatter in the water such as sediments or some phytoplankton types. In our study, we observed that most OCRS algorithms show a strong coupling of $a_{CDOM}(\lambda)^{sat}$ and TSM in all areas of high sediment concentration (compare Fig 7 and 11b).

Our study area, the Laptev Sea shelf, is characterized by extremely shallow waters with frequent resuspension of sediments from the seafloor, for example during storm events. In the Lena River plume, close to the river mouth, where large

amounts of TSM and organic matter are transported to the Arctic Ocean, we expect DOC and TSM to co-vary. Once exported by the Lena River, most particles quickly settle to the seafloor whereas DOM concentration gradually decreases with increasing physical mixing and ongoing degradation. In offshore resuspension areas with very high TSM concentration, DOC and TSM do not necessarily co-vary. Large amounts of terrigenous organic matter can be mineralized on short time scales (about 50 % within a year, Kaiser et al., (2017)) and strongly degraded when deposited in sediments (Bröder et al.,

2016, 2019; Brüchert et al., 2018).

We observe a strong increase of TSM concentrations in the eastern Laptev Sea in September (Fig. 9b) compared to August (Fig. 9a), which is likely caused by differences in wind speed and resulting wave energy leading to resuspension. During acquisitions in August, wind speeds were very low (NCEP reanalysis mean surface wind speed of 2.06 m/s for 75°N & 132.5°E from 01.08.2010 – 05.08.2010) whereas in September winds were stronger (NCEP reanalysis mean surface wind

speed of 6.54 m/s for 75°N & 132.5°E from 04.09.2010 – 20.09.2010). A high TSM concentration in the near-coastal regions around the Lena River Delta, caused by the sediment export by the Lena River, is similar in both mosaics.

The evaluation of OCRS algorithms with in situ data showed the generally good performance of the ONNS and the C2X algorithms (Table 3). However, shallow resuspension areas are not covered by in situ measurements. Thus, the




performance of OCRS algorithms cannot be tested in these areas. Whereas the C2X algorithm derives high $a_{CDOM}(443)$ in the resuspension areas in the eastern Laptev Sea, the ONNS algorithm derives low $a_{CDOM}(440)$ (Fig. 6).

Including all pixels of each scene (Fig. 6), the ONNS-derived $a_{CDOM}(440)$ does not show a linear relationship with TSM concentration (Fig. 10a). However, using only pixels proximal to the Lena River Delta, we observe a correlation

(r=0.68), which is caused by the covariation of TSM and $a_{CDOM}(440)$ in river plume. The C2X-derived $a_{CDOM}(443)$ shows a linear relationship between $a_{CDOM}(443)$ and TSM (r=0.79). The correlation regimes of the $a_{CDOM}(443)$ and TSM from river mouth regions and resuspension areas are visible (Fig. 10b). Thus, we show that C2X-derived $a_{CDOM}(443)$ might vary with TSM. Further confirmation of these satellite-based observations with in situ data is currently not possible due to a lack of in situ data in shallow areas. A partial independence between ONNS-retrieved $a_{CDOM}(440)$ and TSM is of high importance in

shallow Arctic shelf waters, such as the Laptev Sea. Using C2X algorithm, resuspension events would result in erroneous estimation of $a_{CDOM}(443)$.

Furthermore, the C2X-derived TSM concentration is substantially higher compared to TSM concentration derived by ONNS (Fig 12b). Örek et al., (2013) and Heim et al., (2014) report TSM concentrations between 10 and 70 mg/L for Lena River water and up to 18 mg/L in coastal water near the Lena River Delta measured in situ in August 2010. These

15 values are similar to TSM concentrations derived by the ONNS algorithms but lower than C2X algorithm TSM. Considering overestimation of C2X derived $a_{CDOM}(443)$ and TSM compared to in situ data, the use of neural networks trained for a broad range of constituent concentration likely leads to inaccurate results. Combination of neural networks with narrow concentration ranges and a classification into distinct water types (results of classification shown in Fig. C1, Appendix C), as used in ONNS-algorithm, provide more robust and accurate results in regions with a broad range of water types.

**5 Conclusion**

In this study, we demonstrate sources and modification of dissolved organic matter (DOM) by analysing $a_{CDOM}(\lambda)$ characteristics in the fluvial-marine transition zone where the Lena River meets the Laptev Sea. Our results suggest that the $a_{CDOM}(\lambda)$ spectral slope S350-500 could be useful to identify and distinguish processes that degrade DOM at this transition. Comparisons of $a_{CDOM}(\lambda)$ characteristics from this study with reported values from western Arctic waters identify DOM

sources as primarily terrigenous.

We demonstrate the strength of a large in situ dataset that covers multiple water types for deriving the relationship between the optical DOM properties and DOC concentration in surface water of the Laptev Sea and Lena Delta region. The broad range of DOC concentrations and $a_{CDOM}(443)$ from river, coastal and offshore water used to develop this model enables the accurate estimation of DOC by $a_{CDOM}(\lambda)$ in the transition zone between river and seawater. Comparing satellite-

30 retrieved $a_{CDOM}(440)$, using the OCRS ONNS algorithm, and in situ $a_{CDOM}(440)$ demonstrates the performance of the algorithm for these optically complex waters. DOC concentrations calculated from satellite data moderately agreed with in situ DOC measurements ($r^2$=0.53), demonstrating the applicability of the DOC-$a_{CDOM}(\lambda)$ relationship from our compiled

dataset. ONNS-derived $a_{CDOM}(440)$ was found to be independent of the suspended sediment concentration. Thus, resuspension events and resulting sediment-rich backscattering waters seem to have no or little influence on the accuracy of ONNS-derived $a_{CDOM}(440)$.

The Arctic coastal waters of the Laptev Sea are a key region for the fate of terrestrial DOM and can be monitored synoptically using optical remote sensing with a reasonable accuracy. MERIS-retrieved DOC concentrations presented in this study provide a detailed picture of the spatial distribution of the DOC-rich Lena River water on the Laptev Sea shelf and indicate the rapid changes in the magnitude of DOC concentrations in the surface waters within short time periods. If cloud distribution allows, optical remote sensing provides data of high spatial and temporal resolution to track freshwater pathways in the Arctic Ocean, which is of high interest to the oceanographic community.

**Data Availability**

Data has been made available through PANGAEA:

Juhls, Bennet; Hölemann, Jens A; Heim, Birgit; Overduin, Pier Paul; Gonçalves-Araujo, Rafael; Hieronymi, Martin; Fischer, Jürgen (2019): Surface water Dissolved Organic Matter (DOC, CDOM) in the Laptev Sea and Lena River. https://doi.pangaea.de/10.1594/PANGAEA.898813

**Appendix**

**Appendix A**

The regression between DOC and $a_{CDOM}(\lambda)$ was performed for a number of selected wavelengths ($\lambda$) to enable comparisons with other studies. Table A1 shows regression coefficients dependent on wavelengths.

**Appendix B**

Performance of all tested OCRS algorithms are shown in Fig. B1. Whereas ONNS and C2X provide reasonable results close to the 1 to 1 line compared to in situ data, other algorithms (C2R, C2RCC, FUB/WeW) underestimate $a_{CDOM}(\lambda)^{sat}$ strongly.

**Appendix C**

The percentage membership of each pixel is then used to calculate a weighted sum of different neural networks trained for different OWTs. Figure C1 shows the OWTs of the processed scenes from Figure 6 & 7. It is visible that Lena River plume in the coastal waters was classified as OWT 1 (see 1 in Fig. C1) which indicates optically complex, extreme absorbing and high scattering water. The plume between the Lena Delta and the New Siberian Island is characterized by OWT 5 (see 2 in Fig. C1a & b), which indicated a mixture of high absorbing and scattering waters. The Lena River water plume with generally case 2, optically complex waters is sharply delineated to the west, where different water types occur (see 3 in Fig.





C1 & b). Waters west of this plume were classified as OWT 11 displaying Case 1 waters (generally optically deep waters) waters with a small fraction of absorbing waters.

## Author Contributions

B.J., P.P.O. developed the study design. Field work and data collection for this study was conducted by B.J. in all used years, by P.P.O. in 2016 and 2017, J.H. in 2010, 2011, 2013 and 2014 and by B.H. in 2010, 2013, 2014, 2015. M.H. processed satellite data with the ONNS algorithm. B.J. compiled and processed all presented data and prepared the manuscript with editorial contributions from all co-authors.

## Competing interests

The authors declare that they have no conflict of interest

## Acknowledgements

This work was financially supported by Geo.X, the Research Network for Geosciences in Berlin and Potsdam, (grant # SO_087_GeoX). Transdrift expedition data were obtained within the framework of the Laptev Sea System project, supported by the German Federal Ministry of Education and Research (BMBF grant # 03G0833) and the Ministry of Education and Science of the Russian Federation. Part of this study was funded by the Japan Aerospace Exploration Agency (JAXA) GCOM-C project (Contract #: 16RSTK-007867) to AM. We thank the crews and colleagues on-board the research vessels involved in sampling. We are grateful to the colleagues of the Russian-German Otto-Schmidt-Laboratory in St. Petersburg for support and accessibility of laboratory instruments for sample analysis. NCEP-Reanalysis data were provided by the NOAA-CIRES Climate Diagnostics Center, Boulder, CO, USA at http://www. cdc.noaa.gov/. Furthermore, thanks to Antje Eulenburg for laboratory analysis of several parameter datasets. The Horizon 2020 Nunataryuk project (grant # 773421) provided support for discussions with a larger group of experts.

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





# Figures

![Figure 1 map]

**Figure 1: Map of the Laptev Sea and the Lena River Delta region with sample locations from 11 Russian-German expeditions; upper left map shows the Arctic Ocean and the location of the Laptev Sea on the Russian Arctic shelf. Bathymetry is shown by black contour lines and water depth in meters.**




**Figure 2: Boxplot of (a) DOC concentration, (b) a$_{CDOM}$(443), (c) S275-295 and (d) S350-500 for the three water types clustered by salinity (river <0.2, coastal <16, offshore >16); the red line indicates median of each water type.**



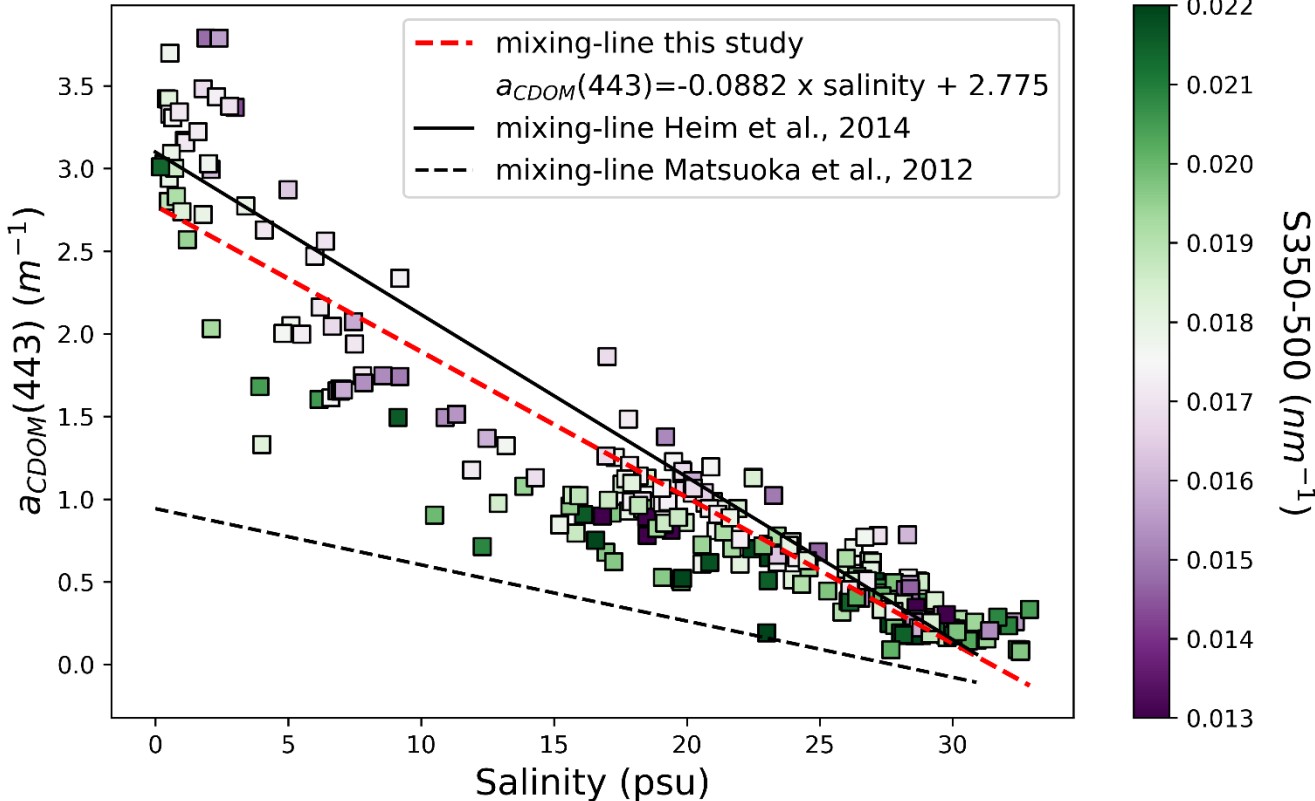

**Figure 3: Relationship between a$_{CDOM}$(443) and salinity (n=283, r²=0.87) for all available water sampled from less than 10 m water depth and a salinity >0.2; color of data-points indicates S350-500; red dashed line shows the linear fit representing the mixing line between Salinity and a$_{CDOM}$(443) within this dataset. Solid black line shows the reported mixing-line from Heim et al., (2014) and**
5   **dashed black line to one from Matsuoka 2012 (adapted to a$_{CDOM}$(443) using Eq. (2) and a constant slope of 0.018.**





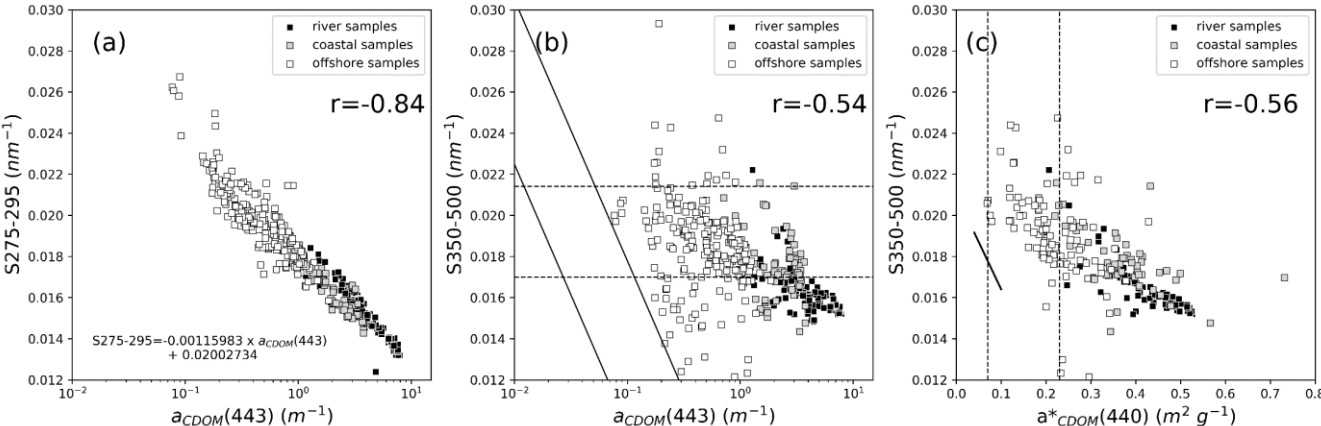

**Figure 4: (a) Relationship between a$_{CDOM}$(443) and S275-295; (b) a$_{CDOM}$(443) vs. S350-500 with 95% confidence intervals of regressions of western Arctic coastal waters (dashed lines) and for western Arctic oceanic water (solid lines) reported by Matsuoka et al., (2011), (2012), (c) a\* a$_{CDOM}$(440) vs. S350-500 with dashed lines representing the borders of a\*$_{CDOM}$(440) for oceanic waters report by Nelson and Siegel, (2002) and solid line shows the reported relationship between a\*$_{CDOM}$(440) and S350-500 from Matsuoka et al., (2012).**





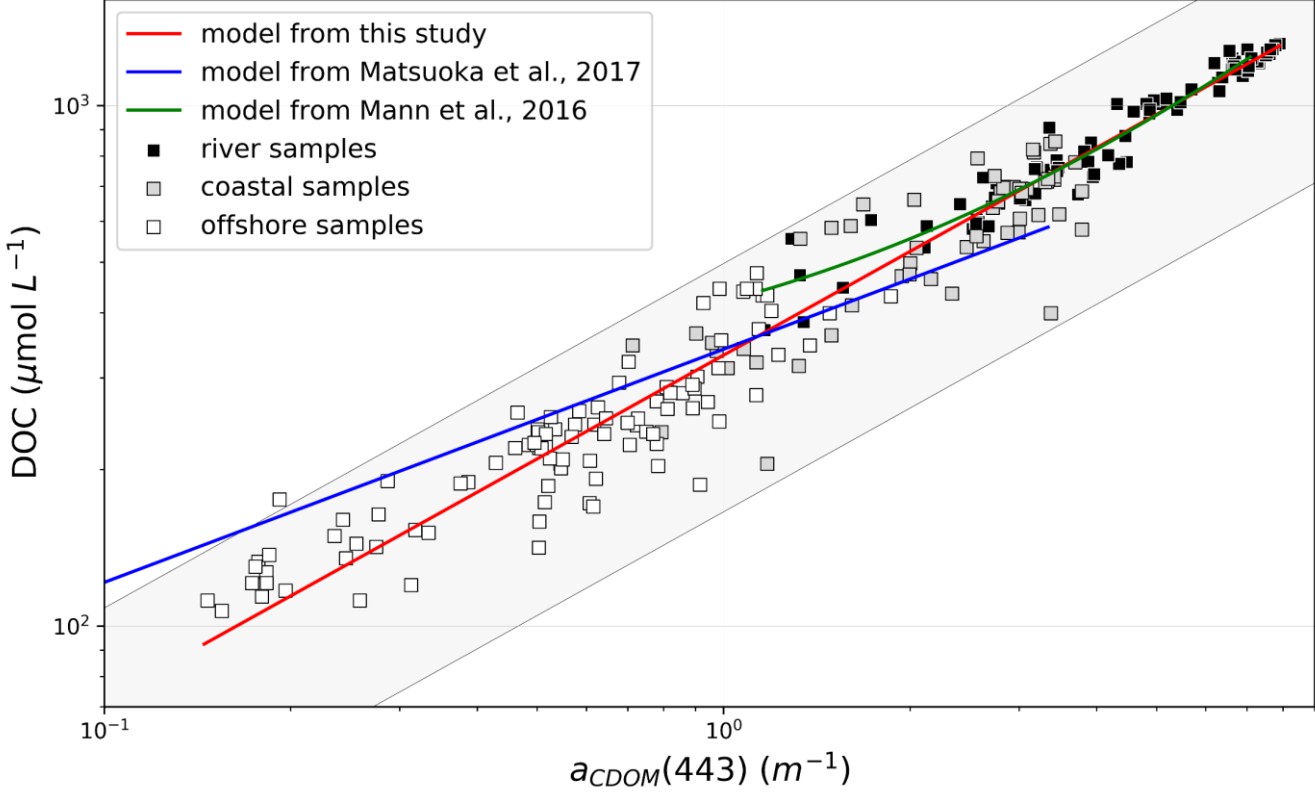

**Figure 5: Relationship between a$_{CDOM}$(443) and DOC (r²=0.96). Red line shows the derived model from this dataset. The blue line shows the relationship from Matsuoka et al. (2017) for a pan-Arctic dataset for offshore and coastal waters. The green line shows the relationship for Lena River water from Mann et al., (2016). The filled grey area shows the 50% error range. Note that the axes are displayed in log-scale.**





**Figure 6 Surface water a$_{CDOM}$(λ)$^{sat}$ from MERIS mosaic from 5 scenes from September 2010 (scenes listed in Table 1) for all tested OCRS algorithms (C2R, ONNS, C2RCC, C2X, FUB/WeW). Squares show in situ a$_{CDOM}$(443) (a$_{CDOM}$(440) for ONNS) with colors according to same color scale as satellite data.**



**Figure 7:** **(a)** Quasi-true color image from 5. August 2010; **(b)** Surface water ONNS-DOC concentration from satellite mosaics from 2010-08-03 to 2010-08-05. **(c)** Quasi-true color image from 7. September 2010. **d)** Surface water ONNS-DOC concentration from satellite mosaics from 2010-09-07 and 2010-09-18 to 2010-09-20. Squares in (b) and (d) squares indicate in situ concentrations with same color scale as satellite data.





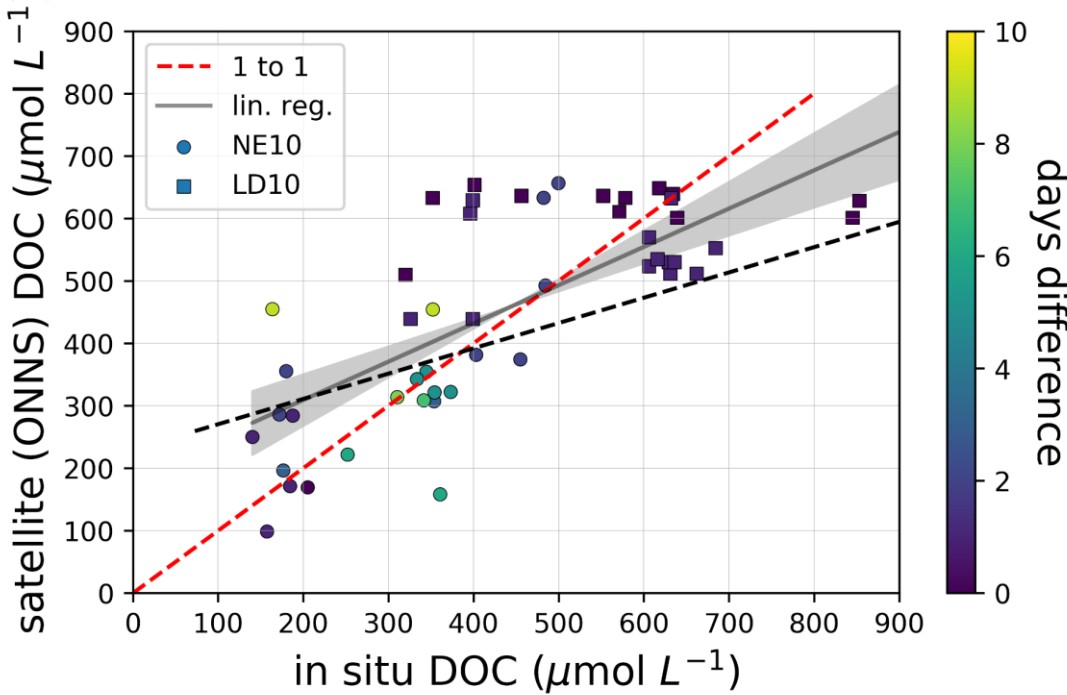

**Figure 8: Comparison of in situ DOC and ONNS-derived DOC.** The dark gray line shows a linear regression (r²=0.53, slope=0.61, n=50). The gray area represents the 95 % confidence interval. The red line indicates 1:1 correspondence. Days difference (symbol color) shows the temporal offset between the satellite scene and in situ water sampling. The dashed black line shows the satellite DOC concentration calculated by using the DOC-$a_{CDOM}$(443) relationship from Matsuoka et al., (2017).




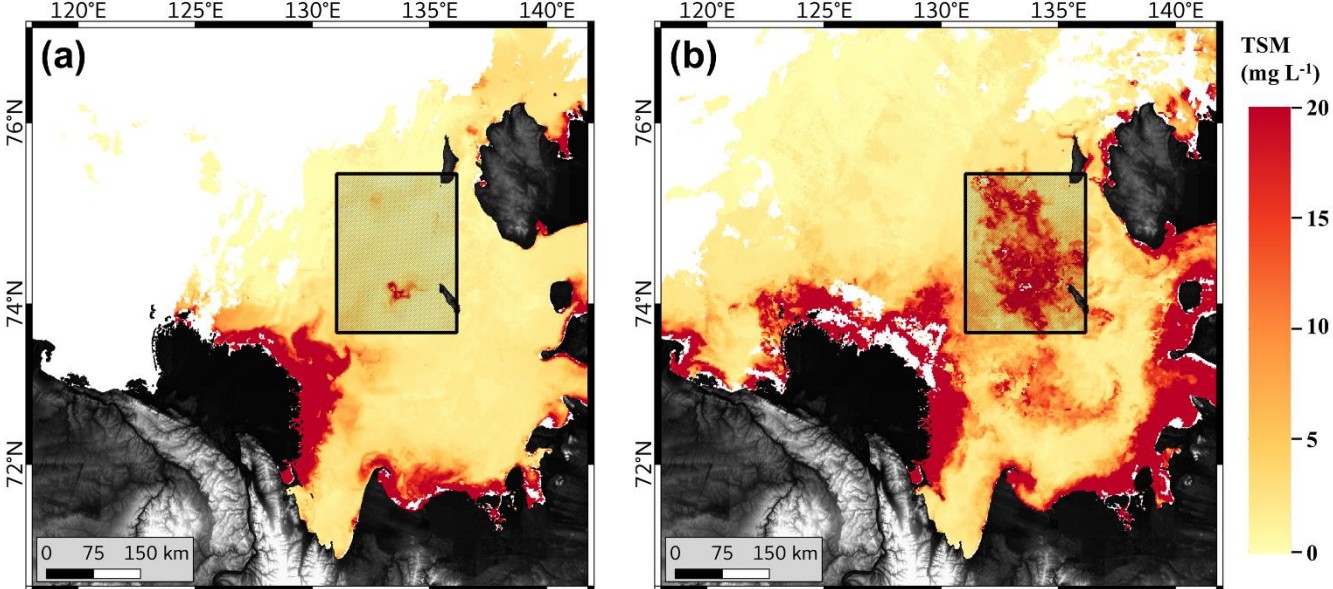

**Figure 9: ONNS-derived TSM concentration for satellite mosaics from (a) August 2010 and (b) September 2010. Shallow water area is highlighted by black square.**




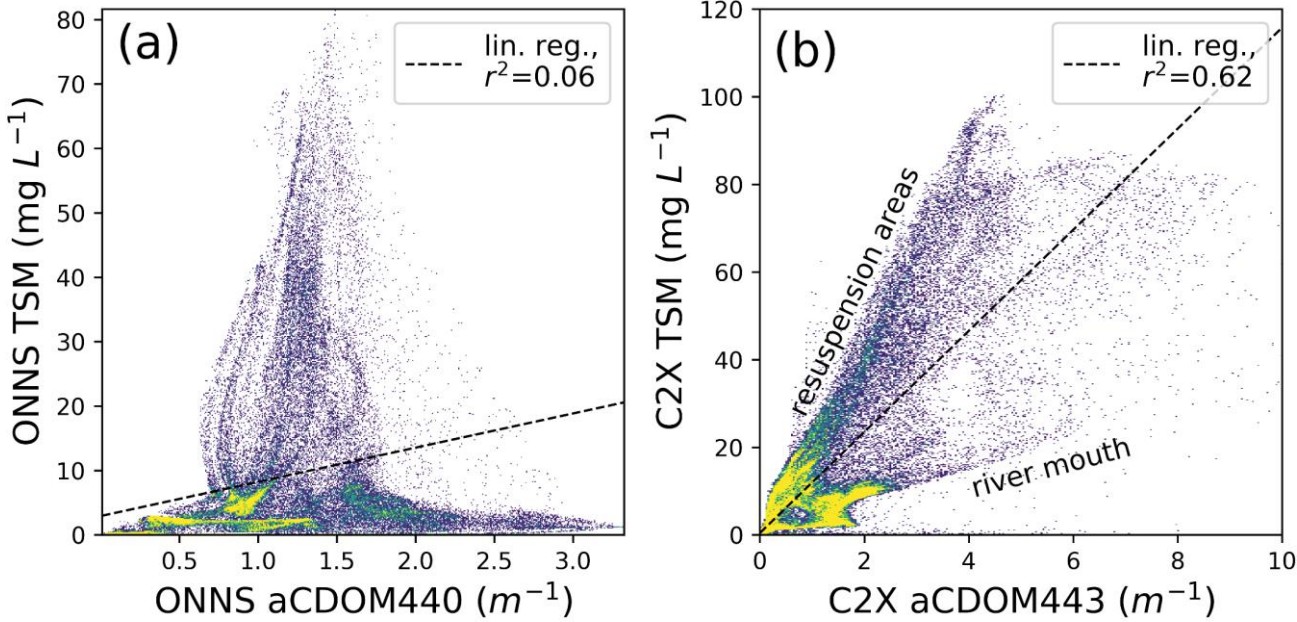

**Figure 10 Relationship between (a) ONNS retrieved a<sub>CDOM</sub>(440) and TSM concentration and (b) C2X retrieved a<sub>CDOM</sub>(440) and TSM concentration for the MERIS scene from 07.09.2010. Relationship, using other scenes from September, is not varying significantly (18.09.2010: ONNS r²=0.22 and C2X r²=0.55, 19.09.2010: ONNS r²=0.23 and C2X r²=0.67, 20.09.2010: ONNS r²=0.03 and C2X r²=0.66)**





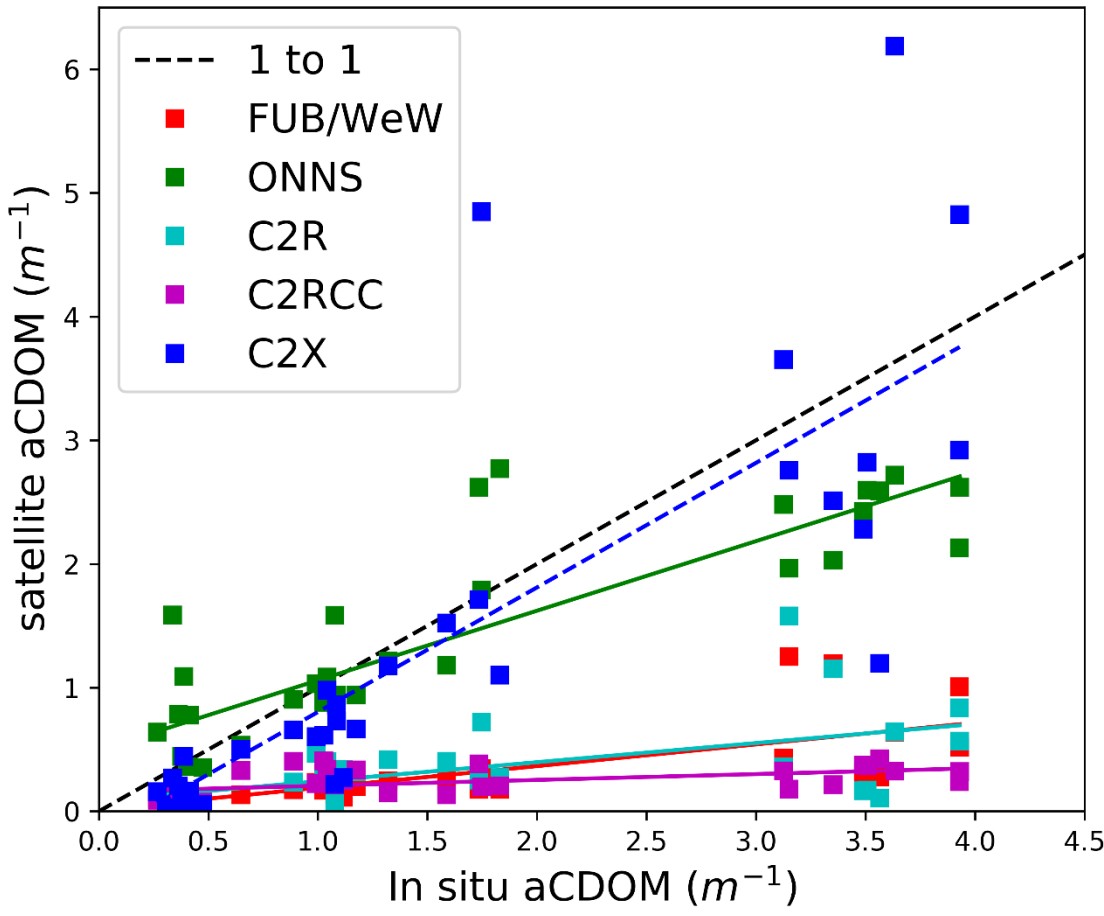

**Figure B1: Comparison of in situ $a_{CDOM}(443)$ or $a_{CDOM}(440)$ with $a_{CDOM}(\lambda)^{sat}$ from different OCRS algorithms.**





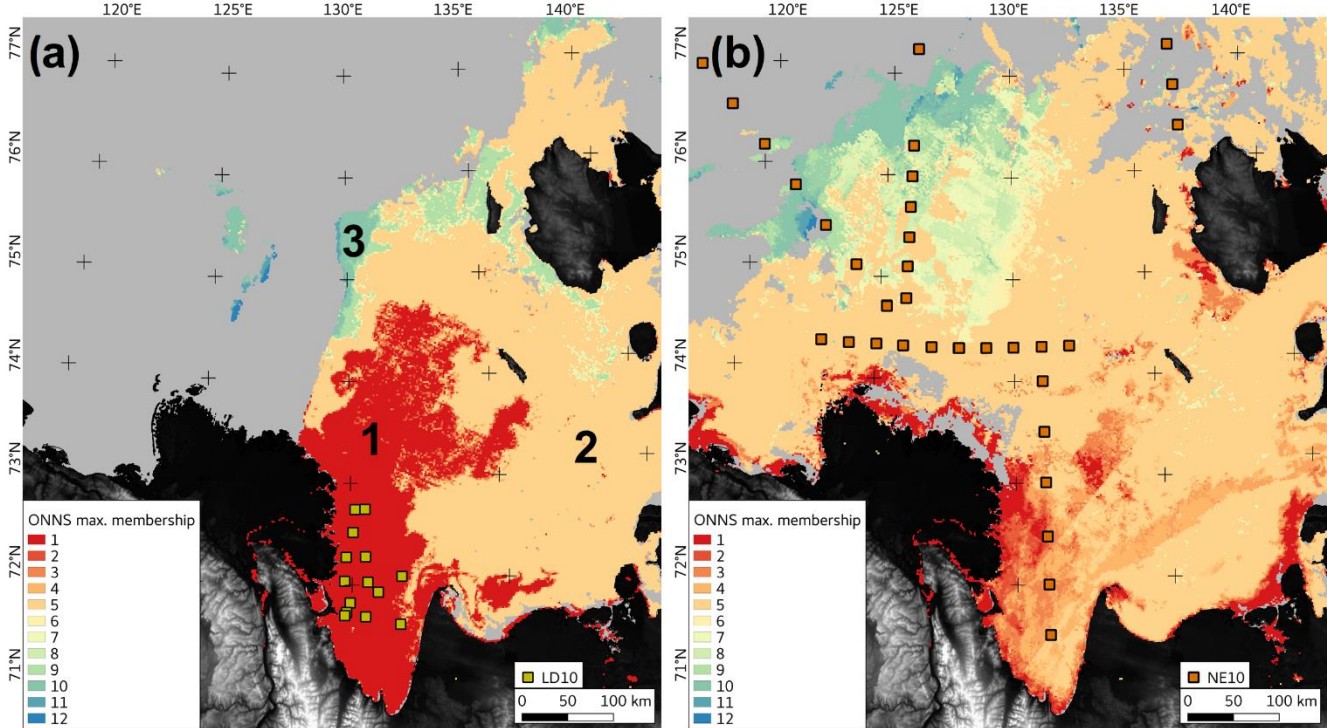

**Figure C1: Optical water types from ONNS fuzzy logic classification for (a) average of 2010-08-03 to 2010-08-05 and (b) average of 2010-09-07 and 2010-09-18 to 2010-09-20**

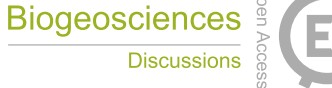



**Tables**

**Table 1: List of MERIS scenes used in this study**

| Scene name | Date, Time (UTC) | Match-up with |
|---|---|---|
| MER_RR__1PRBCM20100803_020534_000005942091_00404_44045_0005 | 2010-08-03 02:05 | LD10 |
| MER_RR__1PRBCM20100804_031401_000005942091_00419_44060_0004 | 2010-08-04 03:14 | LD10 |
| MER_RR__1PRBCM20100805_024241_000005942091_00433_44074_0003 | 2010-08-05 02:42 | LD10 |
| MER_RR__1PRBCM20100907_034618_000005942092_00405_44547_0002 | 2010-09-07 03:46 | NE10 |
| MER_RR__1PRBCM20100918_030140_000005942093_00061_44704_0006 | 2010-09-18 03:01 | NE10 |
| MER_RR__1PRBCM20100919_023010_000005942093_00075_44718_0007 | 2010-09-19 02:30 | NE10 |
| MER_RR__1PRBCM20100920_033916_000005942093_00090_44733_0008 | 2010-09-20 03:39 | NE10 |



**Table 2: Expedition focus regions, years and region. Mean and standard deviation of hydrographic and DOM parameters. The number of samples between DOC and $a_{CDOM}(443)$ differs for some expeditions and "n.a." indicates that no DOC measurements were made.**

| Expedition (Code) | Focus region | Year | Season | S (psu) | DOC ($\mu$mol L$^{-1}$) | $a_{CDOM}(443)$ (m$^{-1}$) | S275-295 (nm$^{-1}$) | S350-500 (nm$^{-1}$) |
|---|---|---|---|---|---|---|---|---|
| Lena 2010 (LD10) | Coastal | 2010 | Aug. | 6.03 ± 6.59 | 563 ± 156 (n=29)[1] | 3.39 ± 0.27 (n=9) | 0.0152 ± 0.0006 | 0.0167 ± 0.0019 |
| Transdrift XVIII (NE10) | Central shelf | 2010 | Sept. | 23.6 ± 6.6 | n.a. | 0.66 ± 0.46 (n=107) | 0.0196 ± 0.0016 | 0.0175 ± 0.0028 |
| Transdrift XIX (YS11) | Central shelf | 2011 | Aug. & Sept. | 19.6 ± 3.6 | 239 ± 55 (n=29) | 0.75 ± 0.21 (n=26) | 0.0193 ± 0.0009 | 0.0161 ± 0.0129 |
| Lena 2013 (LD13) | Lena River | 2013 | July & Aug. | 0.01 ± 0.05 | 695 ± 77 (n=28)[2] | 3.25 ± 0.6 (n=28) | 0.016 ± 0.0007 | 0.0166 ± 0.0006 |
| Gonçalves-Araujo et al., (2015) (GA 13) | Coastal | 2013 | July & Aug. | 14.2 ± 9.4 | 398 ± 155 (n=59)[4] | 1.5 ± 0.86 (n=42)[3] | 0.017 ± 0.0015 | 0.0181 ± 0.00158 |
| Transdrift XXI (VB13) | Central shelf | 2013 | Aug. & Sept. | 22.6 ± 6.9 | n.a. | 0.71 ± 0.55 (n=19) | 0.0201 ± 0.0023 | 0.0184 ± 0.0017 |
| Lena 2014 (LD14) | Lena River | 2014 | May & June | 0.01 ± 0.05 | 1049 ± 248 (n=50) | 5.66 ± 1.85 (n=44) | 0.0145 ± 0.001 | 0.0159 ± 0.0005 |
| Transdrift XXII (VB14) | Central shelf | 2014 | Sept. | 28.3 ± 2.9 | 176 ± 53 (n=46) | 0.36 ± 0.19 (n=47) | 0.0208 ± 0.0021 | 0.0196 ± 0.00164 |
| Lena 2015 (LD15) | Lena River | 2015 | July & Sept. | 0.01 ± 0.05 | n.a. | 2.66 ± 0.72 (n=12)[5] | 0.0167 ± 0.0009 | 0.0166 ± 0.0006 |
| Lena 2016 (LD16) | Lena River & Coastal | 2016 | Aug. & Sept. | 7.3 ± 9.5 | 499 ± 164 (n=17) | 2.47 ± 1.22 (n=35) | 0.0164 ± 0.001 | 0.0164 ± 0.0014 |
| Bykovsky 2017 (Byk17) | Coastal | 2017 | June & July | 1.3 ± 2.2 | 675 ± 61 (n=22) | 2.6 ± 0.69 (n=22) | 0.0161 ± 0.0009 | 0.019 ± 0.0013 |

[1]doi.org/10.1594/PANGAEA.842220

[2]doi.org/10.1594/PANGAEA.844928

5 [3]doi.org/10.1594/PANGAEA.875748

[4]doi.org/10.1594/PANGAEA.842221

[5]doi.org/10.1594/PANGAEA.875754





**Table 3: Performance of tested OCRS algorithms for $a_{CDOM}(\lambda)^{sat}$ with in situ $a_{CDOM}(443)$ or $a_{CDOM}(440)$. Note that not all OCRS algorithms are developed for the highly absorbing waters (high $a_{CDOM}(\lambda)$) found in the Arctic coastal region.**

| OCRS algorithm | n | slope | intercept | r² | Rt | SIQR | %MPE | RMSE |
|---|---|---|---|---|---|---|---|---|
| ONNS | 34 | 0.571 | 0.493 | 0.716 | 0.679 | 0.217 | 58.39 | 0.436 |
| C2R | 34 | 0.157 | 0.087 | 0.329 | 3.23 | 0.075 | 223.38 | 0.277 |
| C2RCC | 34 | 0.048 | 0.157 | 0.271 | 3.23 | 0.093 | 223.05 | 0.097 |
| C2X | 34 | 1.023 | -0.212 | 0.652 | 1.09 | 0.224 | 100.0 | 0.919 |
| FUB/WeW | 34 | 0.178 | 0.013 | 0.545 | 3.76 | 0.057 | 276.49 | 0.2 |





**Table A1: Coefficients selected wavelengths for a_CDOM(λ) using the equation $b * aCDOM(\lambda)^c$**

| λ of a$_{CDOM}$(λ)) | b | c |
|---|---|---|
| 254 | 20.9462548427 | 0.8483590018983822 |
| 350 | 97.4272121688 | 0.7260394049434391 |
| 375 | 136.577758485 | 0.715114349676763 |
| 440 | 322.902097112 | 0.6667788739998305 |
| 443 | 333.695151626 | 0.6640204313768572 |