# Peer review of "Dissolved Organic Matter at the Fluvial-Marine Transition in the Laptev Sea Using in situ Data and Ocean Color Remote Sensing"

_Biogeosciences, 2019_

## Referee Comment (RC1) · Anonymous Referee #1 · 20 Mar 2019

Review of the article:

**Dissolved Organic Matter at the Fluvial-Marine Transition in the Laptev Sea Using in situ Data and Ocean Color Remote Sensing**

The manuscript addresses a relevant scientific issue: sources of dissolved organic matter and carbon cycle in the fluvial-marine transition zone of the Arctic Ocean. The database collected in situ is robust and collaborates to expand knowledge of the sources and processes that occur in the study region, as well as to expand the database. The tests of the various ocean color algorithms show the possibilities of method application and point to certain deficiencies that arise from the available atmospheric correction algorithms dedicated to the satellite image processing.

The estimation of the dissolved organic carbon using satellites images and the empirical algorithm developed by the authors significantly increases the possibility of exploration of ocean color data in the study area, giving the manuscript a substantial contribution to scientific progress within the scope of Biogeosciences.

The method used for DOC estimation was correctly addressed and the results were discussed in an appropriate manner, with figures, tables and graphs presented in a clear, concise and well structured manner. The results are sufficient to support the interpretations and conclusions. The manuscript has the desired scientific quality and follows unrestricted, except minor issues pointed out below.

*Specific Comments:*

**Page 1**

Line 15: inset point "." after "Laptev Sea"

Line 16: change "DOM" to "dissolved organic matter (DOM)"

**Page 2**

Line 2: change "dissolved organic matter (DOM)" to "DOM"

Line 10: change "dissolved organic carbon (DOC)" to "DOC"

Line 24: change "Ocean Color Remote Sensing (OCRS)" to "OCRS"

**Page 3**

Line 31: the first table to be quoted in the paper is table 2, instead of being table 1, please rename the tables so that the first table to be quoted is table 1.

**Page 4**

Line 9: please, see observation above (Page 3 – Line 31)

Line 31: please inform whether the cuvette is quartz or not

**Page 5**

Line 3: please, insert a bibliographical reference that contextualizes the equation 2: "fitting the following equation (bibliographic reference):"

Line 7: change "a*CDOM(440)=aCDOM(440)/DOC)." to "a*CDOM(440)=aCDOM(440)/DOC."

Line 12: after "high spectral resolution and spectroradiometric quality" insert a bibliographical reference

Line 13: Why was not used MERIS Full resolution images (300 m) ???

Line 20: please, explain what was the extraction window dimension (for instance, window of 3 x 3 pixels centered at the sampling points for Rrs median value)

**Page 7**

Line 6: change "and 1.17 to 7.91 m$^{-1}$ aCDOM(443)" to "and 1.17 to 7. 91 m$^{-1}$"

Line 9: change "from to 0.077" to "from 0.077"

Line 18:  change "from river to offshore." to "from river to offshore (Fig. 2)."

Line 25: please, inform what type of correlation (Pearson or  Spearman or ....) was used. It may be better to insert in the paragraph "2.3.1 Functions for satellite retrieval evaluation" a paragraph mentioning the type of correlation used in the statistical analyzes.

**Page 8**

Line 2: the same situation as Line 25, Page 7. Please, see the above comment.

Line 8: change "Nelson and Siegel, (2002) indicating..." to "Nelson and Siegel (2002), dashed lines in Figure 4c, indicating..."

Line 9: please, insert bibliographic reference after "higer aromaticity (bibliographic reference)"

Line 10: change "from (Matsuoka et al., 2012) deviates..." to "from (Matsuoka et al., 2012), solid line in Figure 4c, deviates..."

**Page 9**

Line 18: I didn´t find "Fig.11, A2"

**Page 10**

Line 7: change "...DOC concentrations using..." to "...DOC concentrations (Fig. 8) using..."

Line 8: change "...the presented DOC-a$_{CDOM}$($\lambda$) model (Fig.8) and..." to "...the presented DOC-a$_{CDOM}$($\lambda$) model (Eq. 7) and..."

Line 24: change "…function of salinity, indicating…." To "…function of salinity (Fig 3), indicating..."

Line 25: delete "(Fig. 3)"

**Page 11**

Line 12: please, insert a bibliographic reference here "… which is often used in the OCRS community (bibliographic reference),…"

**Page 13**

Line 14:  this acronym was quoted for the first time "TSM" ??? Please, write out the full text "total suspended matter (TSM) …"

Line 17: I didn´t find "Fig. 11b"

**Page 14**

Line 2:  change "low" to "lower"

Line 13: I didn´t find "Fig 12b"

---

## Referee Comment (RC2) · Piotr Kowalczuk (Referee) · 31 Mar 2019

Piotr Kowalczuk
Institute of Oceanology
Polish Academy of Sciences
Ul. Powstańców Warszawy 55
PL – 81 - 712 Sopot

March 31, 2019

Dr. Gwenaël Abril
Associate Editor,
Biogeosciences

Attn: Review of the manuscript by Bennet Juhls, Pier Paul Overduin, Jens Hölemann, Martin Hieronymi, Atsushi Matsuoka, Birgit Heim, and Jürgen Fischer entitled: "Dissolved Organic Matter at the Fluvial-Marine Transition in the Laptev Sea Using in situ Data and Ocean Color Remote Sensing." submitted to Biogeosciences, , Manuscript No. BG-2019-70

Dear Dr Abril,

After reading the manuscript by Juhls et al., entitled: "Dissolved Organic Matter at the Fluvial-Marine Transition in the Laptev Sea Using in situ Data and Ocean Color Remote Sensing." submitted to Biogeosciences, Manuscript No. BG-2019-70, **I recommend this study for publication in Biogeosciences after minor revision.**

**General opinion**

Arctic Ocean receives 10% of fresh water inflow to the Global Ocean, although its volume is only 1% of the world ocean, likely resulting in an already greater load of CDOM in the Arctic than in other oceans (Stedmon et al., 2011). Majority Large Arctic rivers play an increasingly recognized role in regional carbon cycling by transporting a proportion of terrigenous material from land to the ocean. Significant quantities of dissolved organic matter (DOM) accompany this fresh water flux causing higher than average dissolved organic carbon (DOC) concentrations in the Arctic Ocean relative to other ocean basins (Hernes and Benner,2006; Mann et al., 2016). Combined discharge from six major Arctic rivers (Kolyma, Ob', Lena, Yenisey, Mackenzie and Yukon) constitutes up to 64% of the total fresh water discharge to Arctic Ocean. Therefore monitoring of terrestrial input of DOM from those rivers to the Arctic Ocean is necessary for understanding a carbon cycle in this region. Remoteness and harsh environmental conditions in the region severely reduces applications of routine field monitoring methods in the Arctic. The remote sensing could be a helpful solution, in spite of its regional limitations, as it offer a broad spatial coverage and provides a synoptic picture of many biogeochemical variables. Any new regional Arctic studies providing new algorithms for retrieval biogeochemical variables are very important and have great value.

Authors have presented an empirical model linking optical signatures of dissolved organic matter and dissolved organic carbon concentration, based on large data set of in situ measured CDOM absorption coefficient and DOC concentration collected in multiple expeditions in the Lena river estuary and Laptev Sea. Derived relationship was applied to CDOM absorption coefficients values estimated from ocean color remote sensing data from different processing

algorithms. Authors have compared products from 3 neural network ocean color algorithms applied to MERIS data. The assessment of CDOM absorption coefficient at λ=440 or 443 nm, proved that the ONNS algorithm showed best performance in the extremely CDOM rich water of Lena river estuary and adjacent coastal Laptev Sea waters. The empirical relationship between $a_{CDOM}$(443) and DOC was then applied to OCRS CDOM absorption coefficient retrieval and the surface maps of DOC distribution in the Laptev Sea were produced. The modelled DOC values were compared with DOC measured in situ showing moderate accuracy ($R^2 = 0.53$).

Author have undertaken a challenging task, due to overall difficulty in ocean color remote sensing in the high latitude polar areas due to persistent cloud cover and very low level of upwelling radiance, caused by low Sun zenith angle. Therefore their calibration/validation exercise could not meet rigorous criteria set for such studies by Gregg and Casey (2004). The principle requirement of calibration/validation of ocean color remote sensing is the maximum mismatch between time of in situ observation and time of satellite scene acquisition no longer than 3 hours. This criterion is almost impossible to achieve in Arctic coastal waters, characterized by high heterogeneity of spatial distribution of optically significant constituents, in riverine plume, which distribution is forced by winds, tides and frequent bottom sediments resuspension events in the presence of drift sea ice and high cloudiness. Authors have honestly acknowledged this being very conservative in their assessments of satellite imagery products. Personally I highly acknowledge their results, that have been achieved against all odds.

A manuscript presented me for review contains new and innovative approach to derive spatial and temporal distribution of important biogeochemical variable. I do not have any major critical comments on methodology applied by authors and presentation of their work and results. I have found some minor mistake that could be corrected during revision (listed in the detailed comments section), and I do recommend publication of manuscript by Juhls et al., in Biogeosciences after minor revision.

**Detailed comments**

Abstract

Page 1, Line 15.
Minor technical remark concerning use of optical symbols letter "a" in $a_{CDOM}(\lambda)$ shall be italicized. Please apply this format to all optical symbols in the manuscript.

Page 1 Line 16,

"Observed changes in $a_{CDOM}$ and its …"
Please specify wavelength of CDOM absorption coefficient at which this quantity was measured, referred.

Introduction

Page 2 Line 2,

"Large volumes of fresh water and dissolved organic matter (DOM) are discharged by Arctic rivers into the Arctic Ocean …"

Please give number estimated of fresh water and DOC discharge to Arctic Ocean based on cited literature.

Page 2, Lines 8-9

" …from the Lena River, which delivers around one fifth of all river water to the Arctic Ocean …"

Based on cited literature, please give number estimated of Lena River fresh water discharge.

Page 2 Line 13,

"… the Lena River has the highest peak concentrations of DOC of all Arctic rivers …"

How much is peak DOC concentration – please give a number.

Page 2 Line 31,

"… focused on optically deep (Case 1) waters …"

Wrong citation, Kutser et al., 2017, did not developed the optical classification to optical Case 1 and Case 2 waters. When referring to optical Case 1 water type, please cite original paper by Morel and Prieur, (1977), where this concept was formulated. Alternatively you can cite paper by former students of prof. Andre Morel, who has given an updated interpretation of Case 1, Case 2 water classification, in the paper by Antoine et al., 2014 (Annu. Rev. Mar. Sci. 2014. 6:1–21). The citation to paper by Mobley et al., 2004 is correct.

Page 3 Lines 6 – 14

A paragraph on relationships between $a_{CDOM}(\lambda)$ and DOC. You should mention a paper by Massicotte et al., (2017) who has presented a consistent global relationships between $a_{CDOM}(\lambda)$ and DOC based on more than 12000 in situ measurements from variety of fresh, estuarine, coastal, marine and oceanic environments.

Page 3 Lines 19-21

"Spectral characteristics of $a_{CDOM}(\lambda)$ and their correlation to the DOC specific absorption coefficient …"

I could not find any relationship between spectral characteristics of $a_{CDOM}(\lambda)$ (spectral slope coefficient) and DOC specific absorption coefficient in the paper by Stedmon at al., (2011). I found a table that presented a SUVA(254) values of Siberian and North American Rivers, which indicated slightly lower DOC specific absorption for Yukon River in comparison to Siberian Rivers. The relationship mentioned by Authors has been published by Matsuoka et al., (2012) in the Western Artic Ocean, by Makarewicz et al., (2018) in European section of Arctic Ocean and by Norman et al., (2011) in Antarctica. Most of relationships between those variables were published for tropical/subtropical and temperate estuaries e.g. Mississippi River (Fichot and Benner, 2011, 2012) or Red River Delta, French Guyana and English Channel (Vantrepotte et al. 2015). Please rewrite this sentence specifying exactly which type of spectral characteristics you mean, and refer to paper, where correlation between defined variables could be found.

Material and Methods

Page 5 Line 6

"Spectral slopes of $a_{CDOM}(\lambda)$ were calculated fitting Eq. (2) for the individual wavelength range."

Please, specify which wavelength range you used to calculate spectral slope coefficient $S$.

Page 5 Section 2.3 Satellite data

Please specify which satellite algorithms has been used to derive total suspended matter concentration values that have been used for analysis in Discussion. This information is missing.

Results

Page 8 Line 7-8

"Most samples from this study are located below the $a^*_{CDOM}(440)$ limits of oceanic water reported by Nelson and Siegel, (2002) …"

It would be good if global data distribution of DOC specific CDOM absorption coefficient presented in thepaper by Massicotte at al., (2017) would be used here for comparison.

Page 8 Sectiobn3.3

I think that first paragraph of this section is an introduction to subsection 3.4.

References

Hansell Carlson and Amon,

Wrong citation. I assume that you referred to book edited by Dennis A. Hansell and Craig A. Carlson. I quickly browsed through both editions on line and could not find any chapter authored by Hansell Carlson and Amon. Please give exact bibliographic citation, which chapter and which edition you have cited.

Hieronymi et al 2016 – this is a conference paper, listed as submitted to ESA special publication.  I am not sure if this is a peer reviewed publication and shall be included in references lis.
Please give . Please give exact bibliographic citation of the paper you cite not the link to the web site. If it is a web journal you should use DOI citation.

Best regards,

Piotr Kowalczuk

---

## Author Comment (AC1) · 3 Jun 2019

**Author responses to review of Referee #1 of Biogeosciences manuscript bg-2019-70 "Dissolved Organic Matter at the Fluvial-Marine Transition in the Laptev Sea Using in situ Data and Ocean Color Remote Sensing "**

by Bennet Juhls, Pier Paul Overduin, Jens Hölemann, Martin Hieronymi, Atsushi Matsuoka, Birgit Heim, Jürgen Fischer

We are very grateful to the anonymous reviewer for the detailed and valuable comments on our manuscript. We are confident that the constructive review and suggestions have contributed to improve the paper during our revisions.
Reviewer comments and our responses are presented below.
Reviewer comments are given in *italic font*, our response in blue regular font and the resulting change in the manuscript in *blue italic*.

**Anonymous Referee #1,**

GENERAL COMMENTS:

*"The manuscript addresses a relevant scientific issue: sources of dissolved organic matter and carbon cycle in the fluvial-marine transition zone of the Arctic Ocean. The database collected in situ is robust and collaborates to expand knowledge of the sources and processes that occur in the study region, as well as to expand the database. The tests of the various ocean color algorithms show the possibilities of method application and point to certain deficiencies that arise from the available atmospheric correction algorithms dedicated to the satellite image processing.*
*The estimation of the dissolved organic carbon using satellites images and the empirical algorithm developed by the authors significantly increases the possibility of exploration of ocean color data in the study area, giving the manuscript a substantial contribution to scientific progress within the scope of Biogeosciences.*
*The method used for DOC estimation was correctly addressed and the results were discussed in an appropriate manner, with figures, tables and graphs presented in a clear, concise and well structured manner. The results are sufficient to support the interpretations and conclusions. The manuscript has the desired scientific quality and follows unrestricted, except minor issues pointed out below."*

We are very glad about these positive comments on our manuscript. We revised our manuscript according to the specific comments below.

SPECIFIC COMMENTS:

*"**Page 1** Line 15: inset point "." after "Laptev Sea" "*
This has been changed accordingly in the text.

*"**Page 1** Line 16: change "DOM" to "dissolved organic matter (DOM)""*
This has been changed accordingly in the text.

*"**Page 2** Line 2: change "dissolved organic matter (DOM)" to "DOM""*
This has been changed accordingly in the text.

*"**Page 2** Line 10: change "dissolved organic carbon (DOC)" to "DOC""*
This has been changed accordingly in the text.

*"**Page 2** Line 24: change "Ocean Color Remote Sensing (OCRS)" to "OCRS""*
This has been changed accordingly in the text.

*"**Page 3** Line 31: the first table to be quoted in the paper is table 2, instead of being table 1, please rename the tables so that the first table to be quoted is table 1. "*

Thank you for this comment, the table number 1 and 2 has been changed and all references have been changed accordingly.

**"Page 4** *Line 9: please, see observation above (Page 3 – Line 31)"*
This has been changed (see above).

**"Page 4** *Line 31: please inform whether the cuvette is quartz or not"*
We used quartz cuvettes for our absorbance measurements. We changed the text in the manuscript accordingly:
*"The quartz cuvette length varied depending on the expected absorption in the sampled water"*

**"Page 5** *Line 3: please, insert a bibliographical reference that contextualizes the equation 2: "fitting the following equation (bibliographic reference):" "*
We inserted two references (Jerlov 1968; Bricaud et al. 1981):
*"Spectral slopes of $a_{CDOM}(\lambda)$ were calculated by non-linearly fitting the following equation (Jerlov, 1969; Bricaud et al., 1981)"*

**"Page 5** *Line 7: change "a\*CDOM(440)=aCDOM(440)/DOC)." To "a\*CDOM(440)=aCDOM(440)/DOC." "*
This has been changed accordingly in the text.

**"Page 5** *Line 12: after "high spectral resolution and spectroradiometric quality" insert a bibliographical reference"*
We inserted Delwart et al. (2007) as a reference:
*"For this study, we chose the Medium Resolution Imaging Spectrometer (MERIS) because of its high spectral resolution and spectroradiometric quality (Delwart et al., 2007)"*

**"Page 5** *Line 13: Why was not used MERIS Full resolution images (300 m)???"*
We used MERIS reduced resolution (RR) data because the signal to noise ratio (SNR) of MERIS RR is higher than that of MERIS full resolution (FR) and, thus, the use of RR data for water applications is recommended (Hu et al., 2012). Figure 1 (of this response letter) shows the comparison of MERIS FR (Fig. 1a) and MERIS RR (Fig. 1b) for the region of interest in this study. MERIS FR scene is characterized by a higher noise and thus the risk of extracting noisy pixel is high.

[Figure]

**Figure 1**: (a) DOC concentration retrieved from MERIS FR scene from 3rd August 2010 and (b) from MERIS RR scene.

Furthermore, our study investigates large areas (Laptev Sea shelf) and large-scale features as the Lena River plume (> 300 km). Since no small-scale features are of special interest in this study, we preferred to use the RR data because of its larger coverage and higher quality compared to MERIS FR. The relationship between in situ DOC and ONNS-derived DOC using FR data was slightly stronger ($r^2$=0.68) compared to RR data ($r^2$=0.53).

We added the following explanation to the manuscript:

*"Scenes with reduced resolution were chosen because of their larger extent and thus better coverage of the in situ data stations. Furthermore, Hu et al., (2012) reported a better signal to noise ratio of MERIS reduced resolution compared to MERIS full resolution data and recommended the use of MERIS reduced resolution data."*

We added following information to section 3.5.1:

*"The use of MERIS full resolution data revealed a slightly better performance ($r^2$=0.68, slope=0.77). However, we preferred the use of reduced resolution data due to the reported better quality (Hu et al., 2012)."*

**"Page 5** *Line 20: please, explain what was the extraction window dimension (for instance, window of 3 x 3 pixels centered at the sampling points for Rrs median value)"*

Thank you for pointing out this missing information. We indeed extracted 3 by 3 pixels of each OCRS product following e.g. Müller et al., 2015 and used the median value for the comparison in chapter 3.4 and 3.5.1. We inserted an additional sentence, which provides this information:

*"To compare in situ with satellite data, we used the median of 3 by 3 extracted pixel values from each single processed OCRS image"*

**"Page 7** *Line 6: change "and 1.17 to 7.91 m-1 aCDOM(443)" to "and 1.17 to 7. 91 m-1" "*

This has been changed accordingly in the text.

**"Page 7** *Line 9: change "from to 0.077" to "from 0.077" "*

This has been changed accordingly in the text.

*"**Page 7** Line 18: change "from river to offshore." to "from river to offshore (Fig. 2)." "*
This has been changed accordingly in the text.

*"**Page 7** Line 25: please, inform what type of correlation (Pearson or Spearman or ....) was used. It may be better to insert in the paragraph "2.3.1 Functions for satellite retrieval evaluation" a paragraph mentioning the type of correlation used in the statistical analyzes. "*
In this line we used the coefficient of determination ($r^2$) to show the goodness of the presented linear fit (red dashed line, Fig. 3). We added this information to the manuscript:
*"As in other river-influenced waters, there was a strong linear relationship between $a_{CDOM}(443)$ and salinity ($r^2=0.87$, $n=283$) (Fig. 3), suggesting that physical mixing prevails and plays a role in near-conservative behavior of $a_{CDOM}(\lambda)$.*

**Page 8** *Line 2: the same situation as Line 25, Page 7. Please, see the above comment.*
In cases where we wanted to show the correlation between two variables without fitting a model, we used Pearson correlation coefficient (r) (e.g. Fig 4). We edited the text accordingly:
*"The strongest correlation was observed between $a_{CDOM}(443)$ and the UV slope $S_{275-295}$ (Fig. 4a, Pearson correlation coefficient (r) =-0.84)."*

**Page 8** *Line 8: change "Nelson and Siegel, (2002) indicating…" to "Nelson and Siegel (2002), dashed lines in Figure 4c, indicating…"*
This has been changed accordingly in the text.

**Page 8** *Line 9: please, insert bibliographic reference after "higher aromaticity (bibliographic reference)"*
The $a^*$cdom (DOC normalized absorption) is a parameter similar to SUVA (DOC normalized absorbance). We added references, which use the slope (Granskog et al., 2012; Helms et al., 2008) as well as references, which use SUVA (Weishaar et al., 2003) as an indicator of aromaticity:
*"Most samples from this study are located below the $a^*_{CDOM}(440)$ limits of oceanic water reported by Nelson and Siegel (2002), dashed lines in Figure 4c, indicating that water samples from this study are primarily river influenced with higher aromaticity (Granskog et al., 2012; Helms et al., 2008; Weishaar et al., 2003)"*

**Page 8** *Line 10: change "from (Matsuoka et al., 2012) deviates…" to "from (Matsuoka et al., 2012), solid line in Figure 4c, deviates…"*
This has been changed accordingly in the text.

**Page 9** *Line 18: I didn´t find "Fig.11, A2"*
Thank you for pointing out this mistake. The text has been changed to Fig. B1.

**Page 10** *Line 7: change "...DOC concentrations using..." to "...DOC concentrations (Fig. 8) using..."*
This has been changed accordingly in the text.

**Page 10** *Line 8: change "...the presented DOC-aCDOM($\lambda$) model (Fig.8) and..." to "...the presented DOC-aCDOM($\lambda$) model (Eq. 7) and..."*
This has been changed accordingly in the text.

**Page 10** *Line 24: change "...function of salinity, indicating...." To "...function of salinity (Fig 3), indicating..."*
This has been changed accordingly in the text.

**Page 10** *Line 25: delete "(Fig. 3)"*
This has been changed accordingly in the text.

**Page 11** *Line 12: please, insert a bibliographic reference here "... which is often used in the OCRS community (bibliographic reference),..."*

We added bibliographic references from Babin et al. (2003) and Matsuoka et al. (2011, 2012):
*"Here we use S350-500, which is often used in the OCRS community (Babin et al., 2003; Matsuoka et al., 2011, 2012), instead of S350-400, which is the wavelength range suggested by Helms et al. (2008)."*

**Page 13** *Line 14: this acronym was quoted for the first time "TSM" ??? Please, write out the full text "total suspended matter (TSM) …"*
The acronym was now introduced in section 2.3.

**Page 13** *Line 17: I didn´t find "Fig. 11b"*
Has been changed to "9b"

**Page 14** *Line 2: change "low" to "lower"*
This has been changed accordingly in the text.

**Page 14 Line** *13: I didn´t find "Fig 12b"*
Has been changed to "Fig. 10"

---

## Author Comment (AC2) · 3 Jun 2019

**Author responses to review of Referee Piotr Kowalczuk of Biogeosciences manuscript bg-2019-70 "Dissolved Organic Matter at the Fluvial-Marine Transition in the Laptev Sea Using in situ Data and Ocean Color Remote Sensing "**

by Bennet Juhls, Pier Paul Overduin, Jens Hölemann, Martin Hieronymi, Atsushi Matsuoka, Birgit Heim, Jürgen Fischer

We are very grateful to the reviewer Dr. Piotr Kowalczuk for the detailed and valuable comments on our manuscript. We are confident that the constructive review and suggestions have contributed to improve the paper during our revisions.
Reviewer comments and our responses are presented below.
Reviewer comments are given in *italic font*, our response in blue regular font and the resulting change in the manuscript in *blue italic*.

**Referee Piotr Kowalczuk,**

GENERAL COMMENTS:

*"Arctic Ocean receives 10% of fresh water inflow to the Global Ocean, although its volume is only 1% of the world ocean, likely resulting in an already greater load of CDOM in the Arctic than in other oceans (Stedmon et al., 2011). Majority Large Arctic rivers play an increasingly recognized role in regional carbon cycling by transporting a proportion of terrigenous material from land to the ocean. Significant quantities of dissolved organic matter (DOM) accompany this fresh water flux causing higher than average dissolved organic carbon (DOC) concentrations in the Arctic Ocean relative to other ocean basins (Hernes and Benner,2006; Mann et al., 2016). Combined discharge from six major Arctic rivers (Kolyma, Ob', Lena, Yenisey, Mackenzie and Yukon) constitutes up to 64% of the total fresh water discharge to Arctic Ocean. Therefore, monitoring of terrestrial input of DOM from those rivers to the Arctic Ocean is necessary for understanding a carbon cycle in this region. Remoteness and harsh environmental conditions in the region severely reduces applications of routine field monitoring methods in the Arctic. The remote sensing could be a helpful solution, in spite of its regional limitations, as it offer a broad spatial coverage and provides a synoptic picture of many biogeochemical variables. Any new regional Arctic studies providing new algorithms for retrieval biogeochemical variables are very important and have great value.*
*Authors have presented an empirical model linking optical signatures of dissolved organic matter and dissolved organic carbon concentration, based on large data set of in situ measured CDOM absorption coefficient and DOC concentration collected in multiple expeditions in the Lena river estuary and Laptev Sea. Derived relationship was applied to CDOM absorption coefficients values estimated from ocean color remote sensing data from different processing algorithms. Authors have compared products from 3 neural network ocean color algorithms applied to MERIS data. The assessment of CDOM absorption coefficient at λ=440 or 443 nm, proved that the ONNS algorithm showed best performance in the extremely CDOM rich water of Lena river estuary and adjacent coastal Laptev Sea waters. The empirical relationship between aCDOM(443) and DOC was then applied to OCRS CDOM absorption coefficient retrieval and the surface maps of DOC distribution in the Laptev Sea were produced. The modelled DOC values were compared with DOC measured in situ showing moderate accuracy ($R^2 = 0.53$).*
*Author have undertaken a challenging task, due to overall difficulty in ocean color remote sensing in the high latitude polar areas due to persistent cloud cover and very low level of upwelling radiance, caused by low Sun zenith angle. Therefore, their calibration/validation exercise could not meet rigorous criteria set for such studies by Gregg and Casey (2004). The principle requirement of calibration/validation of ocean color remote sensing is the maximum mismatch between time of in situ observation and time of satellite scene acquisition no longer than 3 hours. This criterion is almost impossible to achieve in Arctic coastal waters, characterized by high heterogeneity of spatial distribution of optically significant constituents, in riverine plume, which distribution is forced by winds, tides and frequent bottom sediments resuspension events in the presence of drift sea ice and high cloudiness. Authors have honestly acknowledged this being very conservative in their*

*assessments of satellite imagery products. Personally I highly acknowledge their results, that have been achieved against all odds.*
*A manuscript presented me for review contains new and innovative approach to derive spatial and temporal distribution of important biogeochemical variable. I do not have any major critical comments on methodology applied by authors and presentation of their work and results. I have found some minor mistake that could be corrected during revision (listed in the detailed comments section), and I do recommend publication of manuscript by Juhls et al., in Biogeosciences after minor revision.”*

Thank you very much for this positive review on our manuscript.

SPECIFIC COMMENTS:

**Abstract**
**“Page 1**, Line 15.: *“Minor technical remark concerning use of optical symbols letter “a” in aCDOM(λ) shall be italicized. Please apply this format to all optical symbols in the manuscript.”*
This has been changed accordingly in the text and figures.

**“Page 1** Line 16: *“Observed changes in aCDOM and its …” Please specify wavelength of CDOM absorption coefficient at which this quantity was measured, referred.”*
In this study we preferably used the wavelengths 443 nm to allow comparisons with other studies (see Fig. 4) and OCRS data. We specified the wavelength in the text:
*“Observed changes in $a_{CDOM}(443)$ and its spectral slopes indicate that DOM is modified by microbial- and photo-degradation.”*

**Introduction**
**“Page 2** Line 2, *“Large volumes of fresh water and dissolved organic matter (DOM) are discharged by Arctic rivers into the Arctic Ocean …”*
*Please give number estimated of fresh water and DOC discharge to Arctic Ocean based on cited literature. “*
We inserted number from recent literature:
*“Large volumes of fresh water ($3588 \pm 257$ $km^3$ $yr^{-1}$, Syed et al., 2007) and dissolved organic matter (DOM) (25–36 Tg C $yr^{-1}$, Raymond et al., 2007) are discharged by Arctic rivers into the Arctic Ocean (Cooper et al., 2005; Dittmar and Kattner, 2003; Stedmon et al., 2011)”*

**“Page 2**, Lines 8-9 *“ …from the Lena River, which delivers around one fifth of all river water to the Arctic Ocean …”*
*Based on cited literature, please give number estimated of Lena River fresh water discharge. “*
We calculated the mean and the standard deviation of the Lena River discharge based on values from Bauch et al. (2013), Fedorova et al. (2013), Stedmon et al. (2011):
*“The Laptev Sea is a wide shelf sea in the eastern Arctic, characterized by fresh surface waters from the Lena River, which delivers around one fifth (~$609.5 \pm 59$ $km^3$ $yr^{-1}$) of all river water to the Arctic Ocean (Bauch et al., 2013; Fedorova et al., 2013; Stedmon et al., 2011).”*

**“Page 2** Line 13, *“… the Lena River has the highest peak concentrations of DOC of all Arctic rivers …” How much is peak DOC concentration – please give a number. “*
We added peak DOC concentration from Stedmon et al., (2011):
*“Moreover, the Lena River has the highest peak concentrations of DOC of up to 1600 μmol/L (Stedmon et al., 2011) of all Arctic rivers.”*

**“Page 2** Line 31: *“… focused on optically deep (Case 1) waters …”*
*Wrong citation, Kutser et al., 2017, did not developed the optical classification to optical Case 1 and Case 2 waters. When referring to optical Case 1 water type, please cite original paper by Morel and Prieur, (1977), where this concept was formulated. Alternatively you can cite paper by former students of prof. Andre Morel, who has given an updated interpretation of Case 1, Case 2 water*

*classification, in the paper by Antoine et al., 2014 (Annu. Rev. Mar. Sci. 2014. 6:1–21). The citation to paper by Mobley et al., 2004 is correct. "*

We deleted Kutser et al. (2017) and replaced it with Morel and Prieur (1977) and Antoine et al. (2014):

*"However, most OCRS retrieval algorithms have focused on optically deep (Case 1) waters, which usually correspond to open ocean where all optical water constituents are coupled to chlorophyll concentration (Antoine et al., 2014; Mobley et al., 2004; Morel and Prieur, 1977)."*

**"Page 3** *Lines 6 – 14: A paragraph on relationships between aCDOM(λ) and DOC. You should mention a paper by Massicotte et al., (2017) who has presented a consistent global relationships between aCDOM(λ) and DOC based on more than 12000 in situ measurements from variety of fresh, estuarine, coastal, marine and oceanic environments. "*

We added Massicotte et al. (2017) as a global model to the manuscript:

*"In order to estimate DOC concentration from $a_{CDOM}(\lambda)$, a number of empirical relationships between in situ DOC and $a_{CDOM}(\lambda)$ for Arctic regions (Fichot and Benner, 2011; Gonçalves-Araujo et al., 2015; Mann et al., 2016; Matsuoka et al., 2012; Örek et al., 2013; Spencer et al., 2009; Walker et al., 2013), as well as global (Massicotte et al., 2017), are presented in recent studies."*

**"Page 3** *Lines 19-21: "Spectral characteristics of aCDOM(λ) and their correlation to the DOC specific absorption coefficient …"*

*I could not find any relationship between spectral characteristics of aCDOM(λ) (spectral slope coefficient) and DOC specific absorption coefficient in the paper by Stedmon at al., (2011). I found a table that presented a SUVA(254) values of Siberian and North American Rivers, which indicated slightly lower DOC specific absorption for Yukon River in comparison to Siberian Rivers. The relationship mentioned by Authors has been published by Matsuoka et al., (2012) in the Western Artic Ocean, by Makarewicz et al., (2018) in European section of Arctic Ocean and by Norman et al., (2011) in Antarctica. Most of relationships between those variables were published for tropical/subtropical and temperate estuaries e.g. Mississippi River (Fichot and Benner, 2011, 2012) or Red River Delta, French Guyana and English Channel (Vantrepotte et al. 2015). Please rewrite this sentence specifying exactly which type of spectral characteristics you mean, and refer to paper, where correlation between defined variables could be found. "*

Thank you for pointing out this imprecisely written sentence. We restructured the sentence and corrected the references as proposed. Unfortunately, the intercomparisons between studies are limited due to the different choices of $a_{CDOM}$ reference wavelength and the missing data publication of complete spectrally resolved $a_{CDOM}$ data. We indicated that in an additional sentence:

*"Recent studies presented $a_{CDOM}(\lambda)$ slopes at different wavelengths ranges and their correlation to the DOC specific absorption coefficient ($a^*_{CDOM}(440)$) at different wavelengths for the Eastern Arctic Ocean (EAO) (Makarewicz et al., 2018: S300-600 to $a^*_{CDOM}(350)$) and the Western Arctic Ocean (WAO) (Matsuoka et al., 2012: S350-500 to $a^*_{CDOM}(440)$). However, direct comparisons of published studies is made difficult by their use of different reference wavelengths."*

**Material and Methods**
**"Page 5** *Line 6: "Spectral slopes of aCDOM(λ) were calculated fitting Eq. (2) for the individual wavelength range."*

*Please, specify which wavelength range you used to calculate spectral slope coefficient S. "*

We added the missing information to the text:

*"Spectral slopes of $a_{CDOM}(\lambda)$ were calculated fitting Eq. (2) for the individual wavelength ranges (275 to 295 nm for S275-295 and 350 to 500 nm for S350-500)."*

**Satellite data**
**"Page 5** *Section 2.3: Please specify which satellite algorithms has been used to derive total suspended matter concentration values that have been used for analysis in Discussion. This information is missing. "*

We added information into this section, however, in the discussion we always indicate the algorithm which was used to derive TSM:

*"All algorithms used in this study use neural networks trained with databases of radiative transfer simulations or in situ measurements or both to invert the satellite signal into a number of inherent optical water properties such as $a_{CDOM}(\lambda)^{sat}$ and concentrations such as total suspended sediment (TSM)."*

**Results**
*"**Page 8** Line 7-8: "Most samples from this study are located below the a\*CDOM(440) limits of oceanic water reported by Nelson and Siegel, (2002) ..."*
*It would be good if global data distribution of DOC specific CDOM absorption coefficient presented in the paper by Massicotte at al., (2017) would be used here for comparison. "*
We agree that a comparison of the CDOM absorption characteristics to the global distribution from Massicotte et al. (2017) is very useful to better classify our samples and put them into a bigger context. We added the data from Massicotte et al. (2017) into the background of Figure 4 and used colors to indicate the salinity of the samples.

[Figure]

**Figure 4:** *(a) Relationship between $a_{CDOM}(443)$ and S275-295; (b) $a_{CDOM}(443)$ vs. S350-500 with 95% confidence intervals of regressions of western Arctic coastal waters (dashed lines) and for western Arctic oceanic water (solid lines) reported by Matsuoka et al., (2011), (2012), (c) a\*CDOM(440) vs. S350-500 with dashed lines representing the borders of a\*CDOM(440) for oceanic waters report by Nelson and Siegel, (2002) and solid line shows the reported relationship between a\*CDOM(440) and S350-500 from Matsuoka et al., (2012). Circles show global data from Massicotte et al., 2017 where colors indicate the salinity.*

We added information to the text:
*"Compared to the global CDOM absorption characteristics from Massicotte et al. (2017) (Fig 4a to c, colored circles), samples from this study are within the range of freshwater influenced samples with lower salinities and clearly differentiate from high saline oceanic waters."*

*"**Page 8** Section 3.3: I think that first paragraph of this section is an introduction to subsection 3.4. "*
We agree that due to the ending of the paragraph it sounded misplaced. We adapted the text so it suits better for subsection 3.3. However, we think that this paragraph, which lists the three steps, should be in the beginning of this subsection to introduce the reader to the following subsections:
*"Generally, retrieval of optical water properties and water constituents such as DOC from satellite data consists of three steps: (1) atmospheric correction of the top of atmosphere radiance to the water-leaving or the in-water reflectance, which is needed as input for the OCRS algorithms, (2) the retrieval of $a_{CDOM}(\lambda)^{sat}$ from the atmospherically-corrected reflectance received by satellite, and (3) if $a_{CDOM}(\lambda)^{sat}$ is retrieved from OCRS, DOC can be calculated using an in situ DOC versus in situ $a_{CDOM}(\lambda)$ relationship. The direct validation and evaluation of different atmospheric corrections (1) is beyond the scope of this study. In the following, we present a new regional DOC- $a_{CDOM}(\lambda)$ relationship (3) from our compiled in situ dataset."*

**References**
*"**Hansell Carlson and Amon,:** Wrong citation. I assume that you referred to book edited by Dennis A. Hansell and Craig A. Carlson. I quickly browsed through both editions on line and could not find any chapter authored by Hansell Carlson and Amon. Please give exact bibliographic citation, which chapter and which edition you have cited. "*
Thank you for pointing out this mistaken citation. We intended to cite the chapter "Chromophoric DOM in the open ocean" from Nelson and Siegel, (2002) in the end of the previous sentence.

We corrected the text and used the correct citation and reference of the chapter in the book of Hansell et al. (2002):
*"Hereinafter, we refer to satellite derived $a_{CDOM}(\lambda)$ as $a_{CDOM}(\lambda)^{sat}$. CDOM absorbs light in the ultraviolet and visible wavelengths (Green and Blough, 1994) and can be used to estimate DOC concentration (Nelson and Siegel, 2002). Thus, OCRS provides an alternative to discrete water sampling (Matsuoka et al., 2017)."*

***"Hieronymi et al 2016****: this is a conference paper, listed as submitted to ESA special publication. I am not sure if this is a peer reviewed publication and shall be included in references list. Please give exact bibliographic citation of the paper you cite not the link to the web site. If it is a web journal you should use DOI citation. "*
We deleted the non-peer-reviewed reference.